# Visual Prompt Tuning in Null Space for Continual Learning

**Yue Lu**[1], **Shizhou Zhang**[1]*, **De Cheng**[2]*, **Yinghui Xing**[1],
**Nannan Wang**[2], **Peng Wang**[1], **Yanning Zhang**[1]
[1] School of Computer Science, Northwestern Polytechnical University, China
[2] School of Telecommunications Engineering, Xidian University, China
zgxd@mail.nwpu.edu.cn, szzhang@nwpu.edu.cn, dcheng@xidian.edu.cn,
xyh_7491@nwpu.edu.cn, nnwang@xidian.edu.cn, peng.wang@nwpu.edu.cn,
ynzhang@nwpu.edu.cn

## Abstract

Existing prompt-tuning methods have demonstrated impressive performances in continual learning (CL), by selecting and updating relevant prompts in the vision-transformer models. On the contrary, this paper aims to learn each task by tuning the prompts in the direction orthogonal to the subspace spanned by previous tasks' features, so as to ensure no interference on tasks that have been learned to overcome catastrophic forgetting in CL. However, different from the orthogonal projection in the traditional CNN architecture, the *prompt gradient orthogonal projection* in the ViT architecture shows completely different and greater challenges, *i.e.*, 1) the high-order and non-linear self-attention operation; 2) the drift of prompt distribution brought by the LayerNorm in the transformer block. Theoretically, we have finally deduced two consistency conditions to achieve the *prompt gradient orthogonal projection*, which provide a theoretical guarantee of eliminating interference on previously learned knowledge via the self-attention mechanism in visual prompt tuning. In practice, an effective null-space-based approximation solution has been proposed to implement the *prompt gradient orthogonal projection*. Extensive experimental results demonstrate the effectiveness of anti-forgetting on four class-incremental benchmarks with diverse pre-trained baseline models, and our approach achieves superior performances to state-of-the-art methods. Our code is available at `https://github.com/zugexiaodui/VPTinNSforCL`.

## 1 Introduction

Continual learning (CL) is crucial for AI models to adapt to the ever-changing environment by learning sequentially arrived data, where the *catastrophic forgetting* is the key challenge [21, 28]. Recently, prompt tuning-based continual learning methods [40, 32, 34, 44, 10, 22, 38, 46, 20, 12, 18] have been attracting increasing attention due to their impressive performances in the CL field. Existing prompt tuning-based works tackle the downstream continual learning problem by selecting and updating relevant prompts, which is encoded with full task-specific knowledge while exploiting the general knowledge of the pre-trained ViTs [40, 39].

On the contrary, this paper aims to learn each task by tuning the prompts in the direction orthogonal to the subspace spanned by previous tasks' features, so as to ensure no interference with tasks that have been learned to overcome *catastrophic forgetting* in CL. It is worth noting that forgetting can be theoretically resolved by gradient orthogonal projection methods [43, 31, 36, 45], which have

---

*Corresponding authors

38th Conference on Neural Information Processing Systems (NeurIPS 2024).

been extensively explored especially when adapting CNN models. Nevertheless, it remains a huge gap to introduce the orthogonal projection-based methods of CNNs to visual prompt tuning due to the following challenges: 1) the high-order and non-linear self-attention operation; 2) the drift of prompt distribution brought by the LayerNorm in the transformer block. For the linear operation in convolution or fully-connected layers, the output features of old tasks can remain unchanged by updating the weights in the orthogonal subspace of previous input features. While for self-attention, three linear transformations are employed on input tokens, followed by high-order and non-linear operations for the self-attention interaction of tokens. It makes the relationship between the update of prompts and the output image tokens much more complex, far exceeding mere linearity.

In this work, we theoretically deduced two consistency conditions to achieve the *prompt gradient orthogonal projection*, which provide a theoretical guarantee of eliminating interference on previously learned knowledge via the self-attention mechanism in visual prompt tuning. To be concrete, we firstly take the full self-attention and LayerNorm into consideration and derive a strict condition for eliminating the interference through a comprehensive analysis of the forward propagation of the ViT layer. Then we further propose to convert the condition of self-attention into its two sufficient conditions, which enables us to address the challenge of high order and nonlinearity. Thirdly, we propose a constraint of invariant prompt distribution that removes the obstacle to the final simplification of the conditions brought by the LayerNorm. The consistency conditions reveal that if the prompt update can be orthogonal to (1) the normalized previous input image tokens projected with the second-order qkv-transformation matrices of the pre-trained model, and (2) the activated attention map generated by image queries and prompt keys, the interference in visual prompt tuning can be eliminated theoretically.

In practice, based on the proposed consistency conditions, an effective null-space-based approximation solution [36] has been proposed to implement the *prompt gradient orthogonal projection*, while the invariant prompt distribution constraint is implemented by incorporating a loss function which penalizes the drifting of prompt distribution over sequential tasks. We validate our Null-Space Projection for Prompts (NSP$^2$) approach on extensive class-incremental benchmarks: 10- and 20-split CIFAR-100, 10-split ImageNet-R [39] and 10-split DomainNet [38], with the sequential fine-tuning VPT and CLIP models as baselines. Our approach brings 4%∼10% improvements in accuracy, and reduces 9%∼17% forgetting, which is superior to state-of-the-art methods.

Our contributions are summarized as follows: (1) We introduce the orthogonal projection into the visual prompt tuning for continual learning, which comprehensively considers the full operations of a transformer layer on the interference problem. (2) Two sufficient consistency conditions for the self-attention and an invariant prompt distribution constraint for LayerNorm are theoretically deduced, based on which an effective null-space-based approximation solution is introduced to implement the prompt gradient orthogonal projection for visual prompt tuning. (3) Extensive experimental results demonstrate the effectiveness of anti-forgetting on four class-incremental benchmarks with diverse pre-trained baseline models, and our approach achieves superior performances to state-of-the-art methods.

## 2 Related Work

**Prompting-Based Approaches:** Most of the prompting-based approaches adopt a two-stage framework [37, 39, 14, 15, 32, 42, 34, 35, 11, 18, 19]: querying a group of prompts for an individual sample and using them to prompt the pre-trained models. For example, L2P [40] first selects a group of prompts from a prompt pool and then feeds them into the ViT. CPrompt [11] proposes to mitigate the gap between training and testing stages to enhance prediction robustness and boost prompt selection accuracy. These approaches essentially focus on acquisition of task-specific prompts tailored to individual samples. There are also several one-stage methods [2, 22, 38, 44, 20] based on prompt tuning. (1) Slowly updating trainable parameters [10, 44]: *e.g.*, LAE [10] updates an offline expert with a large momentum to reduce the change of features. (2) Expandable backbones [46, 20]: *e.g.*, EASE [46] trains a distinct lightweight adapter module for each new task, and designs a semantic mapping to complement the drift of old class prototypes. (3) Enhancing classifiers rather than focusing on learning features [38, 22, 12]: *e.g.*, ESN [38] proposes an anchor-based classifier alignment approach based on energy-based models. As introduced above, these works still lack of a theoretical solution to the interference problem for visual prompt tuning. In our work, we conduct a deep analysis of this problem and provide a theoretical guidance on eliminating the interference.

**Orthogonal Projection-Based Approaches:** Orthogonal projection-based approaches [43, 4, 8, 31, 36, 17, 45] can theoretically eliminate the interference of new tasks on old tasks for linear layers. OWM [43] constructs a projector to find the direction orthogonal to the input space. GPM [31] first projects new gradients to the subspace important to the old tasks and then subtracts the projected components for updating parameters. Adam-NSCL [36] projects the parameter updates to the approximate null space of previous inputs. However, due to the different relationships between parameter updates and outputs in the linear operation and self-attention, the consistency condition used in CNNs is not directly applicable to the prompt tuning in ViTs. In our work, we derive the consistency conditions for the visual prompt tuning, enabling the application of orthogonal projection-based approaches to it, where the null-space projection [36] is adopted in our approach to get an approximate solution efficiently. We notice that a recently emerged work PGP [26] implements GPM [31] to prompt-based frameworks. However, it obtains the same conclusion as that of the linear operation under a simplified attention, which limits its application and performance as compared in the appendix D .

## 3  Preliminaries

**Continual Learning:** In the setting of continual learning, a network $f(\cdot|\mathbf{\Theta})$ with parameters $\mathbf{\Theta}$ is sequentially trained on a stream of disjoint tasks $\{\mathcal{T}_1, \mathcal{T}_2, \cdots, \mathcal{T}_T\}$, where task $\mathcal{T}_t$ is associated with paired data $\{(\mathcal{X}_t^{}, y_t^{})_{i=1}^{|\mathcal{T}_t|}\}$ of size $|\mathcal{T}_t|$. When a task $\mathcal{T}_t$ arrives, the model $f(\cdot|\mathbf{\Theta})$ would be trained for the current task, while the data from previous tasks is unreachable.

**Forward Propagation of Visual Prompt Tuning in ViT Layers:** We describe the forward propagation process of the ViT layer for visual prompt tuning, as illustrated in Figure 1 . Let $\mathbf{X} \in \mathbb{R}^{N \times D}$ and $\mathbf{P} \in \mathbb{R}^{M \times D}$ denote the $N$ input image tokens of a sample (including the pre-trained class token if available) and $M$ prompts, respectively, where $D$ is the dimension of each token. In the ViT layer, only the prompts $\mathbf{P}$ are trainable parameters. The remaining parameters in LayerNorm, qkv-transformations and subsequent MLP introduced below are pre-trained and kept frozen. We use $\mathbf{Z} = [\mathbf{X}; \mathbf{P}] \in \mathbb{R}^{(N+M) \times D}$ to denote the concatenated input tokens. First, they undergo the LayerNorm [1] operation $\mathrm{LN}(\cdot)$:

$$\mathrm{LN}(\mathbf{Z}) = \frac{\mathbf{Z} - \boldsymbol{\mu}_{\mathbf{Z}}}{\boldsymbol{\sigma}_{\mathbf{Z}}} \odot \boldsymbol{\alpha} + \boldsymbol{\beta}, \tag{1}$$

where $\boldsymbol{\mu}_{\mathbf{Z}}, \boldsymbol{\sigma}_{\mathbf{Z}} \in \mathbb{R}^{N+M}, \boldsymbol{\alpha}, \boldsymbol{\beta} \in \mathbb{R}^{D}$. The $\odot$ and division here denote the element-wise (Hadamard) product and division, respectively. Note that the vectors $\boldsymbol{\mu}_{\mathbf{Z}}, \boldsymbol{\sigma}_{\mathbf{Z}}, \boldsymbol{\alpha}$ and $\boldsymbol{\beta}$ are broadcasted to match the matrices of dimensions $(N + M) \times D$, enabling them to carry out operations with $\mathbf{Z}$. Then the normalized tokens are fed into the qkv-transformations:

$$\mathbf{Q}_{\mathbf{Z}} = \mathrm{LN}(\mathbf{Z})\mathbf{W}_q + \boldsymbol{b}_q, \ \mathbf{K}_{\mathbf{Z}} = \mathrm{LN}(\mathbf{Z})\mathbf{W}_k + \boldsymbol{b}_k, \ \mathbf{V}_{\mathbf{Z}} = \mathrm{LN}(\mathbf{Z})\mathbf{W}_v + \boldsymbol{b}_v, \tag{2}$$

where $\mathbf{W}_{\{q,k,v\}} \in \mathbb{R}^{D \times D}$. The vector $\boldsymbol{b}_{\{q,k,v\}} \in \mathbb{R}^{D}$ is broadcasted to a matrix of dimensions $(N + M) \times D$ to facilitate the addition operation. Next is the self-attention:

$$\mathbf{F}_{\mathbf{Z}} = f_{\mathrm{SA}}(\mathbf{Z}) = \mathrm{softmax}(\frac{\mathbf{Q}_{\mathbf{X}}\mathbf{K}_{\mathbf{Z}}^{\top}}{\sqrt{D}})\mathbf{V}_{\mathbf{Z}}, \tag{3}$$

where $\mathbf{Q}_{\mathbf{X}}$ denotes the image tokens serving as queries. Eq. (3) can be expanded as Affinity, softmax (on rows) and Aggregation operations:

$$\begin{cases} \mathbf{A}_{\mathbf{Z}} = f_{\mathrm{aff}}(\mathbf{Q}_{\mathbf{X}}, \mathbf{K}_{\mathbf{Z}}) = \frac{\mathbf{Q}_{\mathbf{X}}\mathbf{K}_{\mathbf{Z}}^{\top}}{\sqrt{D}} = \frac{\mathbf{Q}_{\mathbf{X}}\begin{bmatrix}\mathbf{K}_{\mathbf{X}}^{\top} & \mathbf{K}_{\mathbf{P}}^{\top}\end{bmatrix}}{\sqrt{D}} \in \mathbb{R}^{N \times (N+M)}, & (4) \\[2mm] \mathbf{S}_{\mathbf{Z}} = \mathrm{softmax}(\mathbf{A}_{\mathbf{Z}}) = \mathrm{softmax}(\begin{bmatrix}\mathbf{A}_{\mathbf{X}} \in \mathbb{R}^{N \times N} & \mathbf{A}_{\mathbf{P}} \in \mathbb{R}^{N \times M}\end{bmatrix}) = \begin{bmatrix}\mathbf{S}_{\mathbf{X}} & \mathbf{S}_{\mathbf{P}}\end{bmatrix}, & (5) \\[2mm] \mathbf{F}_{\mathbf{Z}} = f_{\mathrm{agg}}(\mathbf{S}_{\mathbf{Z}}, \mathbf{V}_{\mathbf{Z}}) = \mathbf{S}_{\mathbf{Z}}\mathbf{V}_{\mathbf{Z}} = \begin{bmatrix}\mathbf{S}_{\mathbf{X}} & \mathbf{S}_{\mathbf{P}}\end{bmatrix}\begin{bmatrix}\mathbf{V}_{\mathbf{X}} \\ \mathbf{V}_{\mathbf{P}}\end{bmatrix} \in \mathbb{R}^{N \times D}. & (6) \end{cases}$$

It is worth noting that the rows of the attention map where the prompts serve as queries (*i.e.*, $\mathbf{Q}_{\mathbf{P}}$) do not need to be computed, as formulated in Eq. (4) and illustrated in Figure 1 . The reason is that in VPT-Deep [13], the output prompts of this ViT layer will be replaced with new trainable prompts in the subsequent layer. Omitting $\mathbf{Q}_{\mathbf{P}}$ has no impact on the output image tokens of the ViT layer, as

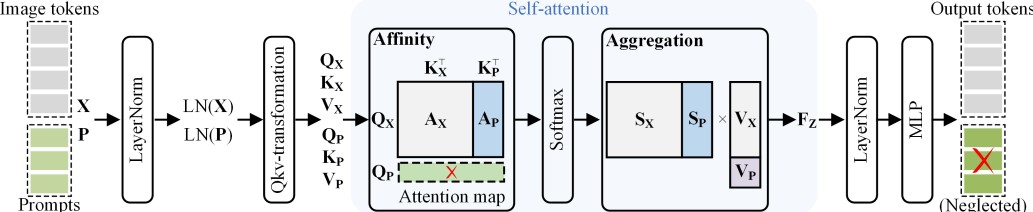

Figure 1: Illustration of the forward propagation in a ViT layer. Residual connections are omitted. The red crosses indicate the rows of attention map or the output prompts can be neglected.

the subsequent Aggregation, LayerNorm and MLP operations are performed independently for each token. If no new prompts are added in the next layer, the output prompts can be just discarded as well.

After the self-attention, operations consist of another LayerNorm and the MLP layer are applied individually to each token, without any interaction among the tokens. Finally, the output fine-tuned image tokens are fed into the next ViT layer.

**Orthogonal Projection in Convolutional Layers:** A convolutional operation is actually a linear operation. For a convolutional layer $f_{\text{conv}}(\cdot|\Theta_t)$ in task $\mathcal{T}_t$, we use $\Theta_t \in \mathbb{R}^{D_{\text{in}} \times D_{\text{out}}}$ to denote its unrolled convolutional kernel matrix [5]. Here, $D_{\text{in}}$ represents the number of pixels within a kernel, and $D_{\text{out}}$ corresponds to the number of kernels. Each convolutional patch from the input feature map is flattened into a row vector with a dimension of $D_{\text{in}}$. These row vectors of totaling $n_p$ patches compose the input feature matrix $\mathbf{X}_t \in \mathbb{R}^{n_p \times D_{\text{in}}}$. The output feature for $\mathbf{X}_t$ in task $\mathcal{T}_t$ is expected to remain unchanged (referred to as consistent) in the next task $\mathcal{T}_{t+1}$ to prevent forgetting:

$$f_{\text{conv}}(\mathbf{X}_t|\Theta_t) = f_{\text{conv}}(\mathbf{X}_t|\Theta_{t+1}). \tag{7}$$

By substituting $\Theta_{t+1} = \Theta_t + \Delta\Theta$, with $\Delta\Theta \neq \mathbf{0}$ denoting the weight update in $\mathcal{T}_{t+1}$, the consistency condition for the convolutional layer is established as follows:

$$\mathbf{X}_t\Theta_t = \mathbf{X}_t(\Theta_t + \Delta\Theta), \tag{8}$$

which can be further simplified as:

$$\mathbf{X}_t\Delta\Theta = \mathbf{0}. \tag{9}$$

Eq. (9) suggests that if the weight update $\Delta\Theta$ is orthogonal to the previous input feature $\mathbf{X}_t$ during training in the new task, the corresponding output feature will remain unchanged. Thereby, the interference of the new task on the old task is eliminated. This can be realized by projecting the candidate weight update $\Theta_{\mathcal{G}}$ into the orthogonal subspace of $\mathbf{X}_t$: $\Delta\Theta = \mathcal{P}\Theta_{\mathcal{G}}$, where $\mathcal{P} \in \mathbb{R}^{D_{\text{in}} \times D_{\text{in}}}$ is an orthogonal projection matrix [43, 36, 31].

Similarly, for the prompt tuning which fine-tunes the prompts $\mathbf{P}_t$ in a ViT layer $f_{\text{vit}}(\mathbf{X}_t|\mathbf{P}_t)$, we also aim to satisfy the following consistency objective for the purpose of anti-forgetting:

$$f_{\text{vit}}(\mathbf{X}_t|\mathbf{P}_t) = f_{\text{vit}}(\mathbf{X}_t|\mathbf{P}_{t+1}). \tag{10}$$

However, the consistency condition in Eq. (9) does not hold for Eq. (10), since $f_{\text{vit}}(\mathbf{X}_t|\mathbf{P}_t) \neq \mathbf{X}_t\mathbf{P}_t$ in prompt tuning. Instead, all the tokens $\mathbf{X}_t$ and $\mathbf{P}_t$ first undergo a LayerNorm and then interact via the self-attention mechanism, as previously described. The complicated forward propagation within the ViT layer brings huge challenge to analyzing the consistency conditions in relation to the prompt update $\Delta\mathbf{P}$. In the next section, we will tackle this challenge and derive the consistency conditions for visual prompt tuning.

## 4 Method

We use $\mathbf{Z}_t = [\mathbf{X}_t; \mathbf{P}_t]$ and $\mathbf{Z}_{t+1} = [\mathbf{X}_t; \mathbf{P}_{t+1}]$ to denote the input tokens before and after updating the prompts, respectively, where $\mathbf{P}_{t+1} = \mathbf{P}_t + \Delta\mathbf{P}, \Delta\mathbf{P} \neq \mathbf{0}$. Our goal is to analyze how to satisfy Eq. (10) and derive one or more conditions expressed in terms of the prompt update $\Delta\mathbf{P}$. These conditions will subsequently guide the application of orthogonal projection to $\Delta\mathbf{P}$.

## 4.1 Analysis of Consistency Conditions

As can be seen in Figure 1, those outputs of LayerNorm and qkv-transformations corresponding to the image tokens remain unaffected by the updates to the prompts. Hence, the essence of attaining the consistency objective Eq. (10) can be turned into analyzing how to keep the output of self-attention in Eq. (3) unchanged as the prompts are updated, *i.e.*, satisfying:

$$\mathbf{F}_{\mathbf{Z}_t} = \mathbf{F}_{\mathbf{Z}_{t+1}}. \tag{11}$$

However, the nonlinear operation (*i.e.*, softmax) and the potential higher-order term $\mathbf{W}_k^\top \mathbf{Z}^\top \mathbf{Z} \mathbf{W}_v$ arising from $\mathbf{K}_{\mathbf{Z}}^\top \mathbf{V}_{\mathbf{Z}}$ in Eq. (3) complicate the direct resolution of this objective. Specifically, the non-injection property of the softmax function causes non-unique solutions. The multiplication between $\mathbf{K}_{\mathbf{Z}_{t+1}^\top} \mathbf{V}_{\mathbf{Z}_{t+1}}$ derives a quadratic term $\text{LN}(\mathbf{P}_t + \Delta \mathbf{P})^\top \text{LN}(\mathbf{P}_t + \Delta \mathbf{P})$, which result in difficult optimization for $\Delta \mathbf{P}$.

To address this issue, we propose two sufficient conditions consisting solely of linear operations. Specifically, we split the process of self-attention into two primary stages, *i.e.*, the Affinity described by Eq. (4) and the Aggregation outlined in Eq. (6). We can achieve Eq. (11) by ensuring the consistency of each stage:

$$\begin{cases} f_{\text{aff}}(\mathbf{Q}_{\mathbf{X}_t}, \mathbf{K}_{\mathbf{Z}_t}) = f_{\text{aff}}(\mathbf{Q}_{\mathbf{X}_t}, \mathbf{K}_{\mathbf{Z}_{t+1}}), & (12) \\ f_{\text{agg}}(\mathbf{S}_{\mathbf{Z}_t}, \mathbf{V}_{\mathbf{Z}_t}) = f_{\text{agg}}(\mathbf{S}_{\mathbf{Z}_{t+1}}, \mathbf{V}_{\mathbf{Z}_{t+1}}). & (13) \end{cases}$$

We first analyze the consistency objective of Affinity, *i.e.*, Eq. (12), for $\mathbf{Z}_t$ and $\mathbf{Z}_{t+1}$:

$$\begin{cases} f_{\text{aff}}(\mathbf{Q}_{\mathbf{X}_t}, \mathbf{K}_{\mathbf{Z}_t}) = \mathbf{Q}_{\mathbf{X}_t} \begin{bmatrix} \mathbf{K}_{\mathbf{X}_t}^\top & \mathbf{K}_{\mathbf{P}_t}^\top \end{bmatrix} = \begin{bmatrix} \mathbf{Q}_{\mathbf{X}_t} \mathbf{K}_{\mathbf{X}_t}^\top & \mathbf{Q}_{\mathbf{X}_t} [\text{LN}(\mathbf{P}_t) \mathbf{W}_k + \boldsymbol{b}_k]^\top \end{bmatrix}, & (14) \\ f_{\text{aff}}(\mathbf{Q}_{\mathbf{X}_t}, \mathbf{K}_{\mathbf{Z}_{t+1}}) = \begin{bmatrix} \mathbf{Q}_{\mathbf{X}_t} \mathbf{K}_{\mathbf{X}_t}^\top & \mathbf{Q}_{\mathbf{X}_t} [\text{LN}(\mathbf{P}_{t+1}) \mathbf{W}_k + \boldsymbol{b}_k]^\top \end{bmatrix}, & (15) \end{cases}$$

where $\sqrt{D}$ is omitted for simplicity. Upon fulfilling Eq. (12), we can obtain $\mathbf{S}_{\mathbf{Z}_t} = \mathbf{S}_{\mathbf{Z}_{t+1}}$, corresponding to the output of Eq. (5). Subsequently, we analyze the consistency objective of Aggregation in Eq. (13), yielding results for $\mathbf{Z}_t$ and $\mathbf{Z}_{t+1}$ as:

$$\begin{cases} f_{\text{agg}}(\mathbf{S}_{\mathbf{Z}_t}, \mathbf{V}_{\mathbf{Z}_t}) = \mathbf{S}_{\mathbf{X}_t} \mathbf{V}_{\mathbf{X}_t} + \mathbf{S}_{\mathbf{P}_t} \mathbf{V}_{\mathbf{P}_t} = \mathbf{S}_{\mathbf{X}_t} \mathbf{V}_{\mathbf{X}_t} + \mathbf{S}_{\mathbf{P}_t} [\text{LN}(\mathbf{P}_t) \mathbf{W}_v + \boldsymbol{b}_v], & (16) \\ f_{\text{agg}}(\mathbf{S}_{\mathbf{Z}_{t+1}}, \mathbf{V}_{\mathbf{Z}_{t+1}}) = f_{\text{agg}}(\mathbf{S}_{\mathbf{Z}_t}, \mathbf{V}_{\mathbf{Z}_{t+1}}) = \mathbf{S}_{\mathbf{X}_t} \mathbf{V}_{\mathbf{X}_t} + \mathbf{S}_{\mathbf{P}_t} [\text{LN}(\mathbf{P}_{t+1}) \mathbf{W}_v + \boldsymbol{b}_v]. & (17) \end{cases}$$

Based on Eq. (12−17), we are able to derive the following two equations, respectively:

$$\begin{cases} \mathbf{Q}_{\mathbf{X}_t} \mathbf{W}_k^\top \text{LN}(\mathbf{P}_t)^\top = \mathbf{Q}_{\mathbf{X}_t} \mathbf{W}_k^\top \text{LN}(\mathbf{P}_{t+1})^\top = \mathbf{Q}_{X_t} \mathbf{W}_k^\top \text{LN}(\mathbf{P}_t + \Delta \mathbf{P})^\top, & (18) \\ \mathbf{S}_{\mathbf{P}_t} \text{LN}(\mathbf{P}_t) \mathbf{W}_v = \mathbf{S}_{\mathbf{P}_t} \text{LN}(\mathbf{P}_{t+1}) \mathbf{W}_v = \mathbf{S}_{\mathbf{P}_t} \text{LN}(\mathbf{P}_t + \Delta \mathbf{P}) \mathbf{W}_v. & (19) \end{cases}$$

Note that we expect to further deduce Eq. (18) and Eq. (19) to obtain equations among $\text{LN}(\mathbf{P}_t)$, $\text{LN}(\mathbf{P}_t + \Delta \mathbf{P})$ and $\Delta \mathbf{P}$. However, due to the square root and quadratic terms in the expressions of the standard deviations $\boldsymbol{\sigma}_{\mathbf{P}_t}$ and $\boldsymbol{\sigma}_{\mathbf{P}_t + \Delta \mathbf{P}}$, it is difficult to express $\boldsymbol{\sigma}_{\mathbf{P}_t + \Delta \mathbf{P}}$ in terms of $\boldsymbol{\sigma}_{\mathbf{P}_t}$ and $\boldsymbol{\sigma}_{\Delta \mathbf{P}}$. Consequently, it is challenging to derive a straightforward equation that relates $\text{LN}(\mathbf{P}_t)$ and $\text{LN}(\mathbf{P}_t + \Delta \mathbf{P})$ through $\Delta \mathbf{P}$.

To simplify the problem, we introduce an additional constraint on the distribution of prompts. Concretely, we require that the updated prompts $\mathbf{P}_t + \Delta \mathbf{P}$ retain the same distribution as $\mathbf{P}_t$, *i.e.*, meeting the following assumption:

$$\begin{cases} \boldsymbol{\mu}_{\mathbf{P}_t + \Delta \mathbf{P}} = \boldsymbol{\mu}_{\mathbf{P}_t}, \\ \boldsymbol{\sigma}_{\mathbf{P}_t + \Delta \mathbf{P}} = \boldsymbol{\sigma}_{\mathbf{P}_t}. \end{cases} \tag{20}$$

In this way, we can establish a straightforward mathematical relationship connecting $\text{LN}(\mathbf{P}_t + \Delta \mathbf{P})$, $\text{LN}(\mathbf{P}_t)$ and $\Delta \mathbf{P}$:

$$\text{LN}(\mathbf{P}_t + \Delta \mathbf{P}) = \frac{\mathbf{P}_t + \Delta \mathbf{P} - \boldsymbol{\mu}_{\mathbf{P}_t + \Delta \mathbf{P}}}{\boldsymbol{\sigma}_{\mathbf{P}_t + \Delta \mathbf{P}}} \odot \boldsymbol{\alpha} + \boldsymbol{\beta} = \frac{\mathbf{P}_t - \boldsymbol{\mu}_{\mathbf{P}_t} + \Delta \mathbf{P}}{\boldsymbol{\sigma}_{\mathbf{P}_t}} \odot \boldsymbol{\alpha} + \boldsymbol{\beta} = \text{LN}(\mathbf{P}_t) + \frac{\Delta \mathbf{P}}{\boldsymbol{\sigma}_{\mathbf{P}_t}} \odot \boldsymbol{\alpha}. \tag{21}$$

Consequently, we can apply Eq. (21) to simplify Eq. (18) and (19) as:

$$
\begin{cases}
\mathbf{Q}_{\mathbf{X}_t}\mathbf{W}_k^\top \mathrm{LN}(\mathbf{P}_t)^\top = \mathbf{Q}_{\mathbf{X}_t}\mathbf{W}_k^\top \mathrm{LN}(\mathbf{P}_t)^\top + \mathbf{Q}_{\mathbf{X}_t}\mathbf{W}_k^\top \Delta\mathbf{P}^\top / \boldsymbol{\sigma}_{\mathbf{P}_t}^\top \odot \boldsymbol{\alpha}^\top, & (22)\\
\mathbf{S}_{\mathbf{P}_t}\mathrm{LN}(\mathbf{P}_t)\mathbf{W}_v = \mathbf{S}_{\mathbf{P}_t}\mathrm{LN}(\mathbf{P}_t)\mathbf{W}_v + \mathbf{S}_{\mathbf{P}_t}\Delta\mathbf{P}\mathbf{W}_v / \boldsymbol{\sigma}_{\mathbf{P}_t} \odot \boldsymbol{\alpha}. & (23)
\end{cases}
$$

It should be noted that in Eq. 22 and Eq. 23, $\mathbf{W}_k$, $\mathbf{W}_v$ and $\boldsymbol{\alpha}$ are pre-trained parameters kept frozen throughout the continual learning process. $\mathbf{Q}_{\mathbf{X}_t}$ and $\mathbf{S}_{\mathbf{P}_t}$ are two matrices derived from the input $\mathbf{X}_t$. As our objective is to ensure that the above two equations remain valid for the variables $\mathbf{Q}_{\mathbf{X}_t}$ and $\mathbf{S}_{\mathbf{P}_t}$, it is sufficient to meet the following conditions, in which $\mathbf{W}_v$ can be ignored whereas $\mathbf{W}_k$ remains crucial:

$$
\begin{cases}
\mathbf{Q}_{\mathbf{X}_t}\mathbf{W}_k^\top \Delta\mathbf{P}^\top = \mathbf{0} & (24)\\
\mathbf{S}_{\mathbf{P}_t}\Delta\mathbf{P} = \mathbf{0} & (25)
\end{cases}
$$

Now we have obtained the simplified formulas expressed by $\Delta\mathbf{P}$ in Eq. (24) and (25).

To sum up, we convert the overall consistency equation Eq. (11) into two sufficient conditions Eq. (12) and (13) for Affinity and Aggregation, respectively. Consequently, we derive two corresponding consistency conditions Eq. (24) and (25) expressed by the prompt update $\Delta\mathbf{P}$, under the constraint of invariant prompt distribution formulated in Eq. (20). The deduced conditions can satisfy the consistency objective in Eq. (10), thereby achieving the goal of eliminating the interference of the new task on the old task for visual prompt tuning.

As $\mathbf{Q}_{\mathbf{X}_t} = \mathrm{LN}(\mathbf{X}_t)\mathbf{W}_q + \boldsymbol{b}_q$, Eq. (24) implies that if the (transposed) prompt update can be orthogonal to the normalized previous input image tokens $\mathbf{X}_t$ projected with a second-order transformation matrices $\mathbf{W}_q\mathbf{W}_k^\top$ of the pre-trained ViT, the consistency for Affinity can be guaranteed. When we ignore the normalization and the bias term in $\mathbf{Q}_{\mathbf{X}_t}$, Eq. (24) can be simplified as $\mathbf{X}_t\mathbf{W}_q\mathbf{W}_k^\top \Delta\mathbf{P}^\top = \mathbf{0}$. The simplified condition is still essentially different from the consistency condition of linear layers (*i.e.*, Eq. (9)) and that deduced in [26] (*i.e.*, $\mathbf{X}_t\Delta\mathbf{P}^\top = \mathbf{0}$). It indicates the interaction between the image tokens and prompts within ViT layers is fundamentally distinct, leading to a unique consistency condition related to the second-order transformation matrices $\mathbf{W}_q\mathbf{W}_k^\top$ of the pre-trained model. Moreover, Eq. (25) is also an essential condition served as one of the sufficient conditions for the consistency of the whole ViT layer. It implies that if the prompt update can be orthogonal to the activated attention map generated by the image queries ($\mathbf{Q}_{\mathbf{X}}$) and prompt keys ($\mathbf{K}_{\mathbf{P}}$), the consistency of Aggregation can be achieved.

## 4.2 Optimization of Consistency Conditions

To jointly optimize Eq. (24) and (25), we need to solve $\Delta\mathbf{P}$ that can meet both equations concurrently. Here, we employ a separate optimization approach to get an approximate solution efficiently. Initially, it ensures $\Delta\mathbf{P}^\top$ is orthogonal to the subspace spanned by $\mathbf{Q}_{\mathbf{X}_t}\mathbf{W}_k^\top$ to satisfy Eq. (24). Subsequently, it makes $\Delta\mathbf{P}$ orthogonal to the subspace spanned by $\mathbf{S}_{\mathbf{P}_t}$ to satisfy Eq. (25).

Specifically, we use $\mathbf{P}_{\mathcal{G}}$ to denote the candidate parameter update generated by the optimizer for the prompts. We aim to obtain a projection matrix $\mathcal{B}$ such that $\Delta\mathbf{P} = \mathcal{B}\mathbf{P}_{\mathcal{G}}$. Following the previously mentioned separate optimization strategy, we first ensure $\Delta\mathbf{P}^\top$ is orthogonal to $\mathbf{Q}_{\mathbf{X}_t}\mathbf{W}_k^\top$ by the projection matrix $\mathcal{B}_1$: $\Delta\mathbf{P}^\top = \mathcal{B}_1\mathbf{P}_{\mathcal{G}}^\top$. Then $\Delta\mathbf{P}$ is made orthogonal to $\mathbf{S}_{\mathbf{P}_t}$ by another projection matrix $\mathcal{B}_2$: $\Delta\mathbf{P} = \mathcal{B}_2\mathbf{P}_{\mathcal{G}}$. Therefore, the objective of the optimization turns into obtaining the two projection matrices $\mathcal{B}_1$ and $\mathcal{B}_2$ to satisfy Eq. (24) and (25). Inspired by the null-space projection method [36], the bases of $\mathcal{B}_1$ and $\mathcal{B}_2$ correspond to the null-space bases of $\mathbf{Q}_{\mathbf{X}_t}\mathbf{W}_k^\top$ and $\mathbf{S}_{\mathbf{P}_t}$, respectively. We use $\mathbf{U}_{1,0} \in \mathbb{R}^{D\times R_1}$ and $\mathbf{U}_{2,0} \times \mathbb{R}^{M\times R_2}$ to denote the bases of the null spaces for $\mathbf{Q}_{\mathbf{X}_t}\mathbf{W}_k^\top$ and $\mathbf{S}_{\mathbf{P}_t}$, where $R_1$ and $R_2$ indicate their nullities. $\mathbf{U}_{1,0}$ and $\mathbf{U}_{2,0}$ can be obtained from the right singular vectors associated with the zero singular values, through the process of singular value decomposition (SVD) applied by $\mathrm{SVD}((\mathbf{Q}_{\mathbf{X}_t}\mathbf{W}_k^\top)^\top \mathbf{Q}_{\mathbf{X}_t}\mathbf{W}_k^\top)$ and $\mathrm{SVD}(\mathbf{S}_{\mathbf{P}_t}^\top \mathbf{S}_{\mathbf{P}_t})$, respectively. In this way, we get the projection matrices $\mathcal{B}_1 = \mathbf{U}_{1,0}\mathbf{U}_{1,0}^\top \in \mathbb{R}^{D\times D}$ and $\mathcal{B}_2 = \mathbf{U}_{2,0}\mathbf{U}_{2,0}^\top \in \mathbb{R}^{M\times M}$, which are the solutions enabling $\Delta\mathbf{P}$ to jointly satisfy Eq. (24) and (25):

$$
\Delta\mathbf{P} = \mathcal{B}_2\mathbf{P}_{\mathcal{G}}\mathcal{B}_1 = (\mathbf{U}_{2,0}\mathbf{U}_{2,0}^\top)\mathbf{P}_{\mathcal{G}}(\mathbf{U}_{1,0}\mathbf{U}_{1,0}^\top). \tag{26}
$$

For the constraint Eq. (20), we incorporate an additional loss function aimed at penalizing the drift of prompt distribution, hence realizing a relaxed version of this constraint:

$$
\mathcal{L}_{\mathrm{LN}} = \|\boldsymbol{\mu}_{\mathbf{P}_{t+1}} - \boldsymbol{\mu}_{\mathbf{P}_t}\|_1 + \|\boldsymbol{\sigma}_{\mathbf{P}_{t+1}} - \boldsymbol{\sigma}_{\mathbf{P}_t}\|_1. \tag{27}
$$

Table 1: Comparison with the baselines ("-Seq") on four benchmarks using two types of models. The upper-bound means jointly training all the classes in the dataset.

| Method | 10S-CIFAR-100 | | 20S-CIFAR-100 | | 10S-ImageNet-R | | 10S-DomainNet | |
|---|---|---|---|---|---|---|---|---|
| | Acc. ↑ | Forgetting ↓ | Acc. ↑ | Forgetting ↓ | Acc. ↑ | Forgetting ↓ | Acc. ↑ | Forgetting ↓ |
| VPT-Seq | 87.27 | 12.33 | 82.36 | 17.36 | 72.46 | 19.41 | 73.28 | 25.65 |
| VPT-NSP$^2$ | **91.74** | **3.28** | **89.89** | **4.91** | **78.88** | **5.06** | **83.54** | **8.54** |
| Upper-bound | 93.87 | - | 93.87 | - | 84.60 | - | 89.25 | - |
| CLIP-Seq | 72.91 | 15.13 | 71.37 | 17.89 | 75.69 | 19.21 | 67.73 | 35.60 |
| CLIP-NSP$^2$ | **80.96** | **12.45** | **79.83** | **13.77** | **82.17** | **6.42** | **77.04** | **18.33** |
| Upper-bound | 84.52 | - | 84.52 | - | 84.86 | - | 81.65 | - |

In Eq. (27), $\boldsymbol{\mu}_{\mathbf{P}_t}$ and $\boldsymbol{\sigma}_{\mathbf{P}_t}$ represent the target prompt distribution obtained in task $\mathcal{T}_t$, while $\boldsymbol{\mu}_{\mathbf{P}_{t+1}}$ and $\boldsymbol{\sigma}_{\mathbf{P}_{t+1}}$ denote the distribution to be optimized in task $\mathcal{T}_{t+1}$.

To sum up, we use Eq. (26) to realize Eq. (24) and (25), and use Eq. (27) to meet Eq. (20), thereby achieving the consistency objective Eq. (10) for anti-forgetting. We provide a full algorithm of our approach in the appendix A .

### 4.3 Extension to Multi-Heads

We further extend the consistency conditions Eq. (24) and (25) to multi-head self-attention, a common feature in current transformer-based models. Suppose there are $H$ heads and $d = D/H$ represents the dimension of each token in a head. We use $\mathbf{Q}_{\mathbf{X}_t.h} \in \mathbb{R}^{N \times d}$, $\mathbf{W}_{k.h} \in \mathbb{R}^{D \times d}$ and $\mathbf{S}_{\mathbf{P}_t.h} \in \mathbb{R}^{N \times M}$ to denote the corresponding matrices in Eq. (24) and (25) for the $h$-th head, respectively. The objective is to ensure these conditions are met across all heads, *i.e.*, $\mathbf{Q}_{\mathbf{X}_t.h}\mathbf{W}_{k.h}^{\top}\Delta\mathbf{P}^{\top} = \mathbf{0}$ and $\mathbf{S}_{\mathbf{P}_t.h}\Delta\mathbf{P} = \mathbf{0}, \forall h \in \{1, 2, \cdots, H\}$. Let $\boldsymbol{\Omega}_{1,t} = [\mathbf{Q}_{\mathbf{X}_t.1}\mathbf{W}_{k.1}^{\top}; \cdots ; \mathbf{Q}_{\mathbf{X}_t.H}\mathbf{W}_{k.H}^{\top}] \in \mathbb{R}^{HN \times D}$ and $\boldsymbol{\Omega}_{2,t} = [\mathbf{S}_{\mathbf{P}_t.1}; \cdots ; \mathbf{S}_{\mathbf{P}_t.H}] \in \mathbb{R}^{HN \times M}$ represent the concatenated matrices from all the heads, respectively. Based on block matrix properties, those two sets of conditions can be formulated as $\boldsymbol{\Omega}_{1,t}\Delta\mathbf{P}^{\top} = \mathbf{0}$ and $\boldsymbol{\Omega}_{2,t}\Delta\mathbf{P} = \mathbf{0}$. To sum up, The main difference between single-head and multi-heads is that the parameter update should be orthogonal to the subspace spanned by the concatenation matrices from all heads for multi-heads self-attention. Therefore, for the multi-heads variant, only an additional step of concatenation of the matrices from all heads is required in our algorithm.

## 5 Experiments

### 5.1 Experimental Setups

In our experiments, we mainly utilize the VPT [13] with a ViT-B/16 backbone [9] pre-trained on ImageNet-21k. Additionally, we validate the effectiveness on the CLIP [27] model, wherein the visual prompts are inserted into the image encoder. Our experiments are conducted across 4 class-incremental benchmarks: 10- and 20-split CIFAR-100, 10-split ImageNet-R and 10-split DomainNet. We report the mean values of the final average accuracy and final average forgetting over 3 runs with different random seeds. Given that the null spaces of $\mathbf{Q}_{\mathbf{X}_t}\mathbf{W}_k^{\top}$ and $\mathbf{S}_{\mathbf{P}_t}$ may not always exist in practice, we compute the approximate null spaces and determine the nullities $R_1$ and $R_2$ in an adaptive manner, rather than the way suggested in [36]. For more detailed information regarding the experimental setups, please refer to Appendix B .

### 5.2 Main Results

**Validation of Effectiveness:** The comparison between our approach and the sequential fine-tuning VPT and CLIP baselines is shown in Table 1 . For the VPT model, the proposed NSP$^2$ achieves 4.47%∼10.26% improvements in accuracy on the 4 benchmarks. Meanwhile, it reduces the forgetting by 9.05%∼17.11%. As to the CLIP model, the NSP$^2$ improves the accuracy by 6.48%∼9.31%, and reduces the forgetting by 2.68%∼17.27%. We calculate the accuracy across all previously encountered tasks after completing training on each task. The accuracy curves of VPT-Seq and VPT-

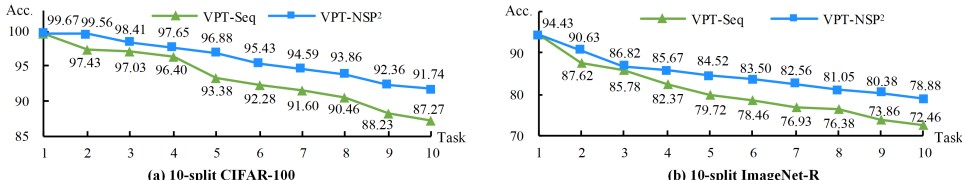

Figure 2: Task-by-task accuracy changing curves of VPT-Seq and VPT-NSP$^2$ on two benchmarks.

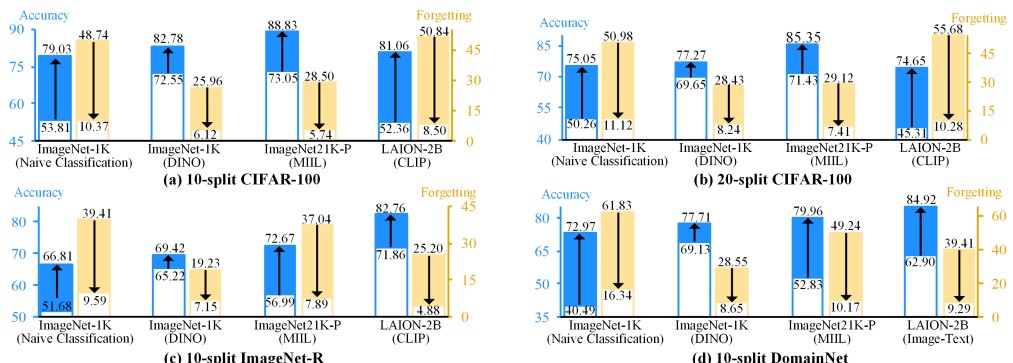

Figure 3: Results of utilizing different pre-training datasets and paradigms. The blue and yellow bars represent accuracy and forgetting, respectively. The upward arrows indicate the accuracy increasing from VPT-Seq to VPT-NSP$^2$, whereas the downward arrows denote the reduction in forgetting.

NSP$^2$ on 10-split CIFAR-100 and 10-split ImageNet-R are displayed in Figure 2. They demonstrate our approach consistently outperforms the baseline throughout the sequential learning of tasks.

We conduct additional experiments with the VPT model, utilizing the weights pre-trained on different datasets as well as different paradigms, as shown in Figure 3. The pre-training paradigms and datasets include: naive classification on ImageNet-1k [30], DINO [3] on ImageNet-1k, MIIL [29] on ImageNet21k-P and CLIP on LAION-2B [6] (we only use its image encoder). As can be seen from the figure, our approach not only significantly enhances accuracy but also markedly mitigates forgetting. These results further demonstrate the generalizability of the proposed approach.

**Comparison with Existing Methods:** We compare our method with existing methods in Table 2, where the competitors include many recent works. The proposed VPT-NSP$^2$ achieves state-of-the-art performance on the four benchmarks, with surpassing the second best approach by an average of 1.49% in accuracy. The forgetting of our approach is not the lowest, which is reasonable since our approach sacrifices some stability for a better trade-off between stability and plasticity. The outperforming accuracy can demonstrate the superiority of our method.

**Ablation Study:** The two consistency conditions Eq. (24) and (25), along with the constraint Eq. (20), constitute the main components of our approach. They correspond to $\mathcal{B}_1$, $\mathcal{B}_2$ in Eq. (26), and $\mathcal{L}_{LN}$ in Eq. (27). We study their effects on the four benchmarks using VPT-NSP$^2$, with results presented in Table 3. We can see that the projection for Affinity ($\mathcal{B}_1$) plays a crucial role, which brings 3.31%∼9.03% improvement in accuracy and 5.42%∼14.76% decline in forgetting. Furthermore, the projection for Aggregation ($\mathcal{B}_2$) and the loss $\mathcal{L}_{LN}$ for invariant prompt distribution are indispensable as well for minimizing forgetting. Optimal accuracy is achieved when all three conditions are applied.

**Model Analysis:** We analyze the evolution of training losses on the 10-split CIFAR-100 and 10-split ImageNet-R benchmarks, as shown in Figure 4. Each point on the curve represents the training loss of the data in $\mathcal{T}_1/\mathcal{T}_2$ after the model has been trained on subsequent tasks. As can be seen, the losses of VPT-NSP$^2$ on previous tasks can be almost retained, confirming that our approach can effectively mitigate the interference of new tasks on old tasks.

**Trade-off between Stability and Plasticity:** We first adaptively determine the nullities $R_1$ and $R_2$ for $\mathcal{B}_1$ and $\mathcal{B}_2$ to achieve near-minimum forgetting. Based on this, we assign two weights $\eta_1$ and $\eta_2$ to the projection matrices to control the trade-off between stability and plasticity: $\Delta \mathbf{P} =$

Table 2: Comparison with existing methods that use the pre-trained ViT-B/16 on ImageNet-21k. The standard deviations are also reported if available. Missing results in the corresponding papers are denoted as "-". The results marked with † and ‡ are implemented by [11] and [10], respectively. The highest accuracies are in bold, and the second highest accuracies are underlined.

| Method | Venue | 10S-CIFAR-100 | | 20S-CIFAR-100 | | 10S-ImageNet-R | | 10S-DomainNet | |
|---|---|---|---|---|---|---|---|---|---|
| | | Acc. | Forgetting | Acc. | Forgetting | Acc. | Forgetting | Acc. | Forgetting |
| L2P [40] | CVPR'22 | $83.83_{\pm0.04}$ | $7.63_{\pm0.30}$ | $80.10_{\pm0.72}$‡ | - | $61.57_{\pm0.66}$ | $9.73_{\pm0.47}$ | $81.17_{\pm0.83}$† | $8.98_{\pm1.25}$ |
| DualPrompt [39] | ECCV'22 | $86.51_{\pm0.33}$ | $5.16_{\pm0.09}$ | $82.02_{\pm0.32}$‡ | - | $68.13_{\pm0.49}$ | $4.68_{\pm0.20}$ | $81.70_{\pm0.78}$† | $8.04_{\pm0.31}$ |
| CODA-P [32] | CVPR'23 | $86.25_{\pm0.74}$ | $1.67_{\pm0.26}$ | - | - | $75.45_{\pm0.56}$ | $1.64_{\pm0.10}$ | $80.04_{\pm0.79}$† | $10.16_{\pm0.35}$ |
| ESN [38] | AAAI'23 | $86.34_{\pm0.52}$ | $4.76_{\pm0.14}$ | $80.56_{\pm0.94}$‡ | - | $62.61_{\pm0.96}$‡ | - | $79.22_{\pm2.04}$† | $10.62_{\pm2.12}$ |
| APG [33] | ICCV'23 | 89.35 | 6.01 | 88.64 | 6.51 | 73.27 | 8.59 | - | - |
| LAE [10] | ICCV'23 | $85.59_{\pm0.46}$ | - | $83.93_{\pm0.28}$ | - | $72.66_{\pm0.63}$ | - | - | - |
| DualP-LGCL [15] | ICCV'23 | $87.23_{\pm0.21}$ | $5.10_{\pm0.15}$ | - | - | $69.46_{\pm0.04}$ | $4.20_{\pm0.06}$ | - | - |
| C-LN [23] | ICCVW'23 | $86.95_{\pm0.37}$ | $6.98_{\pm0.43}$ | - | - | $76.36_{\pm0.51}$ | $8.31_{\pm1.28}$ | - | - |
| EvoPrompt [18] | AAAI'24 | $87.97_{\pm0.30}$ | $2.60_{\pm0.42}$ | $84.64_{\pm0.14}$ | $3.98_{\pm0.24}$ | $76.83_{\pm0.08}$ | $2.78_{\pm0.06}$ | $79.50_{\pm0.29}$ | $3.81_{\pm0.36}$ |
| OVOR-Deep [12] | ICLR'24 | $85.99_{\pm0.89}$ | $6.42_{\pm2.03}$ | $84.13_{\pm0.75}$ | $6.81_{\pm0.77}$ | $76.11_{\pm0.21}$ | $7.16_{\pm0.34}$ | $79.61_{\pm0.86}$ | $4.77_{\pm0.94}$ |
| DualP-PGP [26] | ICLR'24 | $86.92_{\pm0.05}$ | $5.35_{\pm0.19}$ | $83.74_{\pm0.01}$ | $7.91_{\pm0.15}$ | $69.34_{\pm0.05}$ | $4.53_{\pm0.04}$ | $80.41_{\pm0.25}$ | $8.39_{\pm0.18}$ |
| InfLoRA [20] | CVPR'24 | $87.06_{\pm0.25}$ | $6.22_{\pm0.39}$ | $81.42_{\pm0.54}$ | $6.42_{\pm0.33}$ | $75.65_{\pm0.14}$ | $5.73_{\pm0.44}$ | $81.45_{\pm0.68}$ | $5.35_{\pm0.52}$ |
| EASE [46] | CVPR'24 | 87.76 | 5.94 | 85.80 | 7.19 | 76.17 | 7.82 | 78.89 | 7.89 |
| CPrompt [11] | CVPR'24 | $87.82_{\pm0.21}$ | $5.06_{\pm0.50}$ | $83.97_{\pm0.31}$ | $6.85_{\pm0.43}$ | $77.14_{\pm0.11}$ | $5.97_{\pm0.68}$ | $82.97_{\pm0.34}$ | $7.45_{\pm0.93}$ |
| VPT-NSP² | This work | $\mathbf{91.74}_{\pm0.63}$ | $3.28_{\pm0.45}$ | $\mathbf{89.89}_{\pm0.72}$ | $4.91_{\pm0.59}$ | $\mathbf{78.88}_{\pm0.50}$ | $5.06_{\pm0.26}$ | $\mathbf{83.54}_{\pm0.77}$ | $8.54_{\pm0.48}$ |

Table 3: Ablation studies of each component in our approach on the four benchmarks.

| $\mathcal{B}_1$ | $\mathcal{B}_2$ | $\mathcal{L}_{LN}$ | 10S-CIFAR-100 | | 20S-CIFAR-100 | | 10S-ImageNet-R | | 10S-DomainNet | |
|---|---|---|---|---|---|---|---|---|---|---|
| | | | Acc. ↑ | Forgetting ↓ | Acc. ↑ | Forgetting ↓ | Acc. ↑ | Forgetting ↓ | Acc. ↑ | Forgetting ↓ |
| | | | 87.27 | 12.33 | 82.36 | 17.36 | 72.46 | 19.41 | 73.28 | 25.65 |
| √ | | | 90.58 | 6.91 | 88.13 | 10.27 | 78.05 | 8.14 | 82.31 | 10.89 |
| | √ | | 88.74 | 10.85 | 83.32 | 16.48 | 74.71 | 14.69 | 78.87 | 17.81 |
| √ | √ | | 91.33 | 4.22 | 88.96 | 6.42 | 78.37 | 6.25 | 83.17 | 8.95 |
| √ | | √ | 91.42 | 3.94 | 88.46 | 8.64 | 78.30 | 6.31 | 83.13 | 9.32 |
| | √ | √ | 89.36 | 9.32 | 86.67 | 11.59 | 75.27 | 13.35 | 79.45 | 16.50 |
| √ | √ | √ | **91.74** | **3.28** | **89.89** | **4.91** | **78.88** | **5.06** | **83.54** | **8.54** |

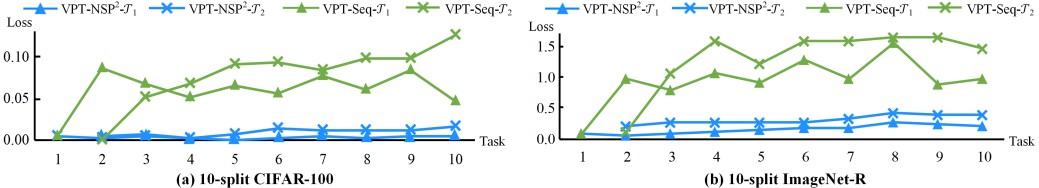

(a) 10-split CIFAR-100    (b) 10-split ImageNet-R

Figure 4: Training loss curves of VPT-NSP² and VPT-Seq on tasks $\mathcal{T}_1$ and $\mathcal{T}_2$ when the models are trained on sequential tasks.

$[\eta_2\mathcal{B}_2 + (1-\eta_2)\mathbf{I}]\,\mathbf{P}_{\mathcal{G}}\,[\eta_1\mathcal{B}_1 + (1-\eta_1)\mathbf{I}]$, where $\mathbf{I}$ denotes the identity matrix. The effects of $\eta_1$ and $\eta_2$ which are set to a same value $\bar{\eta}$ is shown in Figure 5. As the weight decreases, the accuracy increases first owing to better plasticity, and then decreases due to worse stability caused by the forgetting. It implies that a trade-off can be achieved by the two weights of projections.

**Long-sequence Continual Learning** We experiment on 5 benchmarks under the protocols of 50 tasks and 100 tasks to validate that our approach remains effective even within the context of long-sequence continual learning. The results are presented in Table 4. Despite lacking plasticity enhancement, VPT-NSP² can outperform existing state-of-the-art approaches and especially surpasses L2P by a large margin. This demonstrates that forgetting is still the predominant factor affecting performance in long sequence of tasks. With the plasticity enhancement, VPT-NSP² achieves significant increase in accuracy (by 1.1%~2.9%). This demonstrates that our plasticity enhancement is effective in learning new knowledge in long-sequence continual learning.

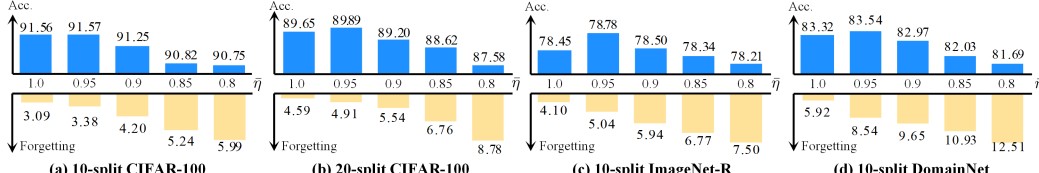

Figure 5: Effect of the projection matrix weight $\bar{\eta}$ on the accuracy and forgetting for the stability-plasticity trade-off on the four benchmarks.

Table 4: Results for 50 tasks and 100 tasks on CIFAR-100, ImageNet-R and DomainNet datasets. † indicates no plasticity enhancement, and ‡ indicates using the balanced plasticity enhancement where $\bar{\eta}$ is the default value less than 1. Our approach still outperforms other methods in long sequences of tasks.

| Method | 50S-CIFAR100 | | 50S-ImageNet-R | | 50S-DomainNet | | 100S-ImageNet-R | | 100S-DomainNet | |
|---|---|---|---|---|---|---|---|---|---|---|
| | Acc. | Forgetting | Acc. | Forgetting | Acc. | Forgetting | Acc. | Forgetting | Acc. | Forgetting |
| L2P | 76.19 | 12.06 | 48.53 | 12.99 | 59.45 | 11.53 | 38.87 | 15.26 | 50.52 | 17.66 |
| EvoPrompt | 76.60 | 13.86 | 68.53 | 10.03 | 67.68 | 10.41 | 61.84 | 15.84 | 56.35 | 21.39 |
| OVOR | 65.69 | 14.28 | 60.08 | 5.86 | 66.27 | 7.43 | 40.49 | 8.12 | 47.65 | 8.91 |
| InfLoRA | 61.49 | 13.68 | 59.02 | 11.02 | 69.96 | 9.51 | 38.16 | 15.11 | 44.32 | 17.85 |
| EASE | 74.47 | 9.31 | 68.17 | 7.76 | 61.20 | 10.01 | 47.55 | 8.22 | 33.08 | 32.14 |
| CPrompt | 74.97 | 7.45 | 68.47 | 8.16 | 67.87 | 9.36 | 56.95 | 10.20 | 53.73 | 12.14 |
| VPT-Seq | 70.47 | 29.21 | 56.38 | 37.91 | 58.39 | 44.79 | 49.72 | 45.53 | 46.39 | 49.34 |
| VPT-NSP[2]† | 81.92 | 6.56 | 67.32 | 6.35 | 70.13 | 9.92 | 59.97 | 10.07 | 54.44 | 11.04 |
| VPT-NSP[2]‡ | **82.98** | 6.66 | **69.48** | 6.51 | **71.28** | 11.36 | **62.23** | 12.13 | **57.35** | 13.82 |

# 6 Conclusion

In this paper, we study the interference problem of visual prompt tuning in ViTs, and propose two consistency conditions which can eliminate the interference in theory under the constraint of invariant prompt distribution. They guarantee the consistency of Affinity, Aggregation and distribution of prompts in LayerNorm, respectively, which jointly achieve the consistency objective of the whole ViT layer. We adopt the null-space projection to implement the two conditions and utilize an extra loss to satisfy the constraint. Our experiments on various benchmarks demonstrate the effectiveness of the proposed conditions for anti-forgetting, and our approach achieves state-of-the-art performances.

**Limitation Discussion:** To simplify the derivation of our consistency conditions, we introduce a constraint of invariant prompt distribution. Although the superior results show that it may not be a very strong assumption, it is not an exact solution.

# Acknowledgments

This work was supported in part by the National Natural Science Foundation of China (NSFC) under Grant 62101453, 62176198 and 62201467; in part by the Project funded by China Postdoctoral Science Foundation under Grant 2022TQ0260 and Grant 2023M742842, in part by the Young Talent Fund of Xi'an Association for Science and Technology under Grant 959202313088, in part by Innovation Capability Support Program of Shaanxi (Program No. 2024ZC-KJXX-043) and in part by the Natural Science Basic Research Program of Shaanxi Province (No. 2022JC-DW-08).

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

# Appendix: Visual Prompt Tuning in Null Space for Continual Learning

## A    Algorithm

An overview and algorithm of our approach are provided in Figure 6 and Algorithm 1 , respectively. We first initialize the overall uncentered covariance matrices [36] $\mathbf{C}_1$ and $\mathbf{C}_2$, as well as the null-space projection matrices $\mathcal{B}_1$ and $\mathcal{B}_2$. During training, the cross-entropy loss for classification and the loss of prompt distribution $\mathcal{L}_{\text{LN}}$ are jointly utilized for optimization. Subsequently, we get the candidate prompt updates $\mathbf{P}_{\mathcal{G}}$ computed by the optimizer. Then $\mathbf{P}_{\mathcal{G}}$ is projected by the null-space projection matrices $\mathcal{B}_1$ and $\mathcal{B}_2$ for updating the prompts. After the convergence, we obtain the matrices $\mathbf{J}_1$ and $\mathbf{J}_2$ to temporarily store $\mathbf{Q}_{\mathbf{X}_t}\mathbf{W}_k^{\top}$ and $\mathbf{S}_{\mathbf{P}_t}$ for the data of the current task. Then they are used to update the uncentered covariance matrices $\mathbf{C}_1$ and $\mathbf{C}_2$ by addition. Finally, we update the null-space projection matrices using the uncentered covariance matrices, which will be used in the next task.

Algorithm 2 shows the process of computing a null-space projection matrix. First, an input uncentered covariance matrix $\mathbf{C}$ is decomposed by SVD, from which we can get the singular values and right singular vectors. Next, we determine the nullity $R$ (*i.e.*, the dimension of null space) of $\mathbf{C}$ according to the maximum second derivative, which is introduced in Section C . Then we select $R$ right singular vectors corresponding to the $R$ smallest singular values considered close to 0 as the bases of null space. Finally, we compute the normalized projection matrix, which provides an upper bound for the scale of the projected gradients and prevents excessive gradient magnitudes. In our implementation, the null-space projection matrix is added by an identity matrix with a weight $\eta$ (specifically $\eta_1$ for $\mathcal{B}_1$ and $\eta_2$ for $\mathcal{B}_2$). $\eta$ is a hyper-parameter for the trade-off between stability and plasticity, which is also introduced in Section C

## B    Experimental Setups and Implementation Details

**Models:** We validate our approach on the Vision Transformer (ViT) [9] and CLIP [27] models in the experiments, whose backbones are both ViT-Base/16 [9]. The ViT is pre-trained on ImageNet-21k, and we insert 4 prompts into each of the 12 layers for fine-tuning, which is referred to as "VPT" [13]. The classifiers are dependently trained in each task and the orthogonal projection is not applicable to them. All the classifiers from the available tasks are concatenated to make prediction during inference. For the CLIP model pre-trained on the WebImageText, we insert 4 prompts into each of the first 3 layers of the image encoder, while the text encoder is kept frozen. The logit scale that serves as a learnable scalar parameter to scale the cosine similarities between image features and text features is also set to trainable. We observed a serious cross-task confusion among the tasks in the CLIP model. Hence, we follow [44] to utilize the class-wise mean and covariance of previous features extracted before the embedding projection head (*i.e.*, the last linear layer of the image encoder) to refine the projection head, after the prompt tuning stage in each task.

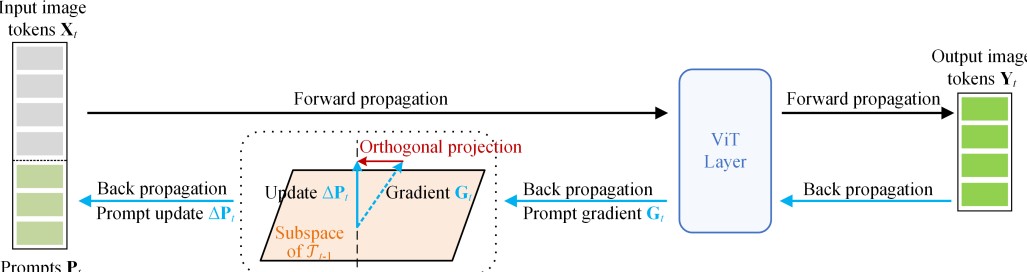

Figure 6: Illustration of our algorithm. The input image tokens with prompts are fed into the ViT layer for forward propagation. During optimization, the gradients of the prompts will be projected into the orthogonal direction to the subspace of the previous task $\mathcal{T}_{t-1}$. The projected prompt update will be used to update the prompts for anti-forgetting.

---

**Algorithm 1** $\text{NSP}^2$ for Visual Prompt Tuning

---

**Inputs:** Datasets $\mathcal{D}_t = \{(\mathcal{X}_t^{}, y_t^{})\}_{i=1}^{|\mathcal{T}_t|}$ for task $\mathcal{T}_t \in \{\mathcal{T}_1, \mathcal{T}_2, \cdots\}$, ViT model $f_{\text{model}}(\cdot|\mathbf{P}_t)$ with the prompts $\mathbf{P}_t$ to be optimized (the classifier is omitted for simplicity), uncentered covariance matrices $\mathbf{C}_1$ and $\mathbf{C}_2$, projection matrices $\mathcal{B}_1$ and $\mathcal{B}_2$

**Outputs:** The optimized prompts $\mathbf{P}_t$

1: **Initialization:** Randomly initialize $\mathbf{P}_t$; $\mathbf{C}_1 = \mathbf{0}$, $\mathbf{C}_2 = \mathbf{0}$, $\mathcal{B}_1 = \mathbf{I}$, $\mathcal{B}_2 = \mathbf{I}$
2: **for** task $\mathcal{T}_t \in \{\mathcal{T}_1, \mathcal{T}_2, \cdots\}$ **do**
3:     **repeat**
4:         Sample a mini-batch $\mathcal{X}_t, \boldsymbol{y}_t \sim \mathcal{D}_t$
5:         Obtain prediction by $\hat{\boldsymbol{y}}_t \leftarrow f_{\text{model}}(\mathcal{X}_t|\mathbf{P}_t)$
6:         Compute the classification loss $\mathcal{L}_{total} \leftarrow \text{CrossEntropy}(\hat{\boldsymbol{y}}_t, \boldsymbol{y}_t)$
7:         **if** $t > 1$ **then**
8:             Compute the loss of prompt distribution $\mathcal{L}_{\text{LN}}$ by Eq. (27)
9:             Accumulate the losses $\mathcal{L}_{total} \leftarrow \mathcal{L}_{total} + \mathcal{L}_{\text{LN}}$
10:         **end if**
11:         Get the candidate prompt update $\mathbf{P}_{\mathcal{G}}$ from the optimizer by the loss $\mathcal{L}_{total}$
12:         **if** $t > 1$ **then**
13:             Compute the prompt update $\Delta\mathbf{P} \leftarrow \mathcal{B}_2 \mathbf{P}_{\mathcal{G}} \mathcal{B}_1$ by the null-space projection Eq. (26)
14:         **else**
15:             Directly adopt the candidate prompt update $\Delta\mathbf{P} \leftarrow \mathbf{P}_{\mathcal{G}}$
16:         **end if**
17:         Update the prompts by $\mathbf{P}_t \leftarrow \mathbf{P}_t - learning\_rate \times \Delta\mathbf{P}$
18:     **until** convergence
19:     Initialize two temporary matrices $\mathbf{J}_1 = [\ ]$ and $\mathbf{J}_2 = [\ ]$
20:     **for** $\mathcal{X}_t^{} \in \mathcal{D}_t$ **do**
21:         Get the matrices $(\mathbf{Q}_{\mathbf{X}_t} \mathbf{W}_k^\top)^{}$ and $\mathbf{S}_{\mathbf{P}_t}^{}$ by the forward propagation $f_{\text{model}}(\mathcal{X}_t^{}|\mathbf{P}_t)$
22:         Update $\mathbf{J}_1$ and $\mathbf{J}_2$ by concatenating $(\mathbf{Q}_{\mathbf{X}_t} \mathbf{W}_k^\top)^{}$ and $\mathbf{J}_1$, $\mathbf{S}_{\mathbf{P}_t}^{}$ and $\mathbf{J}_2$, respectively
23:     **end for**
24:     Update the uncentered covariance matrices $\mathbf{C}_1 \leftarrow \mathbf{C}_1 + \mathbf{J}_1^\top \mathbf{J}_1$ and $\mathbf{C}_2 \leftarrow \mathbf{C}_2 + \mathbf{J}_2^\top \mathbf{J}_2$
25:     Compute the null-space projection matrices $\mathcal{B}_1$ and $\mathcal{B}_2$ by Algorithm 2 using $\mathbf{C}_1$ and $\mathbf{C}_2$
26: **end for**

---

---

**Algorithm 2** Computing Null-Space Projection Matrix

---

**Inputs:** Uncentered covariance matrix $\mathbf{C}$, hyper-parameter $\eta \in [0, 1]$ for the trade-off between stability and plasticity (mentioned in Section C)

**Outputs:** Null-space projection matrix $\mathcal{B}$

1: Get the singular values $\Lambda$ in descending order and the corresponding right singular vectors $\mathbf{U}$ by singular value decomposition $\Lambda, \mathbf{U}^\top \leftarrow \text{SVD}(\mathbf{C})$, where the left singular vectors are omitted
2: Calculate the nullity $R$ by the maximum second derivative as introduced in Eq. (28)
3: Select the right singular vectors of the $R$ smallest singular values in $\mathbf{U}$ as $\mathbf{U}_0 \leftarrow \mathbf{U}_{[D-R:D]}$
4: Compute the projection matrix $\mathcal{B} \leftarrow \frac{\mathbf{U}_0 \mathbf{U}_0^\top}{\|\mathbf{U}_0 \mathbf{U}_0^\top\|_{\text{F}}}$
5: Update $\mathcal{B}$ with the weight $\eta$ by $\mathcal{B} \leftarrow \eta\mathcal{B} + (1 - \eta)\mathbf{I}$ (corresponding to Eq. (29))

---

**Benchmarks:** We conduct experiments under the class-incremental learning protocol, where the classes in each task are disjoint, and task identity is unknown during inference. Four class-incremental benchmarks with three widely used datasets are adopted: 10- and 20-split CIFAR-100, 10-split ImageNet-R [39] and 10-split DomainNet [25, 38]. For the CIFAR-100 dataset, the total of 100 classes are randomly split into 10 or 20 tasks, which can evaluate the ability to handle different numbers of tasks. We follow [39] to randomly split the 200 classes in ImageNet-R into 10 tasks, which forms the 10-split ImageNet-R benchmark. For the 10-split DomainNet, we follow the same dataset protocol adopted in [38] and [11] to select the top 200 classes with the most images from the original DomainNet [25], and randomly split them into 10 tasks with 20 classes per task. 25% samples of the training data in each dataset are picked as a validation set for searching optimal hyper-parameters.

**Metrics:** Formally, the final average accuracy and final average forgetting are defined as:

$$\text{Final average accuracy} = \frac{1}{T}\sum_{i=1}^{T} a_{T,i},$$

$$\text{Final average forgetting} = \frac{1}{T-1}\sum_{i=1}^{T-1} \max_{j\in\{1,2,\cdots,T-1\}}(a_{j,i} - a_{T,i}),$$

where $T$ is the number of tasks, $a_{T,i}$ is the accuracy of the $T$-th model on the $i$-th task samples, and $a_{j,i}$ is the accuracy of the $j$-th model on the $i$-th task samples.

Higher accuracy means the model performs better, while lower forgetting means stronger stability (*i.e.*, the ability to retain old knowledge). However, lower forgetting does not always generate higher accuracy since the accuracy is also affected by plasticity (*i.e.*, the ability to learn new knowledge). The accuracy is the main metric we should focus on as it reflects the precision of classification in practice.

**Implementations Details:** For all the datasets and models, the images fed into the models are resized to $224 \times 224$ pixels and augmented by AutoAugment [7] during training. For the VPT-based models, we use the Adam optimizer [16] with $\beta_1 = 0.9$, $\beta_2 = 0.999$ and a weight decay of $5 \times 10^{-5}$ to train 100 epochs with an initial learning rate of 0.01 and a batch size of 256 on all benchmarks. The learning rate is scaled by a factor of 0.1 at the 50-th and 80-th epoch. Our training losses consist of the cross-entropy loss for classification and the loss $\mathcal{L}_{\text{LN}}$ in Eq. (27) whose coefficient is set to 1. Through cross validation on the validation set, we set the temperatures in the cross-entropy loss to 28, 25, 30 and 30 for the 10-split CIFAR100, 20-split CIFAR100, 10-split ImageNet-R and 10-split DomainNet benchmarks. There are two hyper-parameters $\eta_1$ and $\eta_2$ used for the trade-off between stability and plasticity in null-space projection as introduced in Section C, and we set both of them to be 0.97, 0.95, 0.94 and 0.95 for the four benchmarks by cross validation.

As to the CLIP-based models, the differences in training settings are as follows. We train them for 20 epochs with the batch size of 220 and the learning rate 0.001 which decays at the 10-th and 16-th epoch. The temperatures are all set to 1 since the logit scale is trainable. $\eta_1$ and $\eta_2$ are set to 0.98 which is a proper value for all the benchmarks. We refine the embedding projection head for 50 epochs using the SGD optimizer with a learning rate of 0.001, a momentum of 0.9 and a weight decay of $1 \times 10^{-4}$.

We implement our approach in PyTorch [24] with the timm library [41]. The experiments are performed on a server with 128 GB RAM and four NVIDIA RTX 4090 GPUs. Each of the experiment can be finished in three hours.

## C Trade-off between Stability and Plasticity

Given that the null space of covariance matrix does not always exist in practice, Wang *et al.* [36] suggest approximating it by selecting the bases whose associated singular values approach zero, where the singular values smaller than a specified multiple (denoted as $\gamma$ in our paper) of the smallest one are selected. However, we experimentally find this strategy and the experience for selecting $\gamma$ are not suitable for prompt tuning in ViTs to determine the nullities $R_1$ and $R_2$ for the uncentered covariance matrices $\mathbf{C}_1$ and $\mathbf{C}_2$ in Algorithm 1, which will be introduced afterwards. To solve this problem, we propose an adaptive nullity strategy to determine the nullities in an adaptive manner. Utilizing the characteristic that the curve of descending singular values forms an "L" shape, we divide the curve into two parts by the point where the gradient changes fastest to cover most of the small singular values. It is realized by calculating the maximum second derivative of the points:

$$\begin{cases} R_1 = D - \arg\max_{j}\{\lambda_{j-1} - 2\lambda_j + \lambda_{j+1}\}_{j=2}^{D-1}, \\ R_2 = M - \arg\max_{j}\{\lambda_{j-1} - 2\lambda_j + \lambda_{j+1}\}_{j=2}^{M-1}, \end{cases} \tag{28}$$

where $\lambda_j$ denotes the $j$-th singular value. We find it reaches near-minimum forgetting in our experiments which also means reaching near-optimal stability. Furthermore, to enhance the plasticity, we fuse the projection matrices with identity matrices by the weights $\eta_1 \in [0,1]$ and $\eta_2 \in [0,1]$ which should be close to 1:

$$\Delta\mathbf{P} = [\eta_2\mathcal{B}_2 + (1-\eta_2)\mathbf{I}]\,\mathbf{P}_\mathcal{G}\,[\eta_1\mathcal{B}_1 + (1-\eta_1)\mathbf{I}]. \tag{29}$$

In this way, we can make a trade-off between stability and plasticity by enhancing the plasticity based on near-optimal stability, and $\eta_1$ and $\eta_2$ are the hyper-parameters to control the trade-off.

## D  Comparison with PGP

### D.1  Difference in Methods

The main difference between our method and PGP [26] are summarized as follows. (1) We derive a different consistency condition for Affinity even if we ignore the LayerNorm operation and the bias terms in the qkv-transformation. Specifically, our simplified consistency condition for Affinity is $\mathbf{X}_t \mathbf{W}_q \mathbf{W}_k^\top \Delta \mathbf{P}^\top = \mathbf{0}$, contrasted with $\mathbf{X}_t \Delta \mathbf{P}^\top = \mathbf{0}$ in PGP. (2) We analyze the consistency conditions for the complete self-attention, *i.e.*, $\mathrm{softmax}(\frac{\mathbf{Q_x K_Z^\top}}{\sqrt{D}})\mathbf{V_Z}$ which contains the Aggregation operation. However, PGP does not account for the Aggregation. (3) We take the LayerNorm before self-attention into consideration and propose an invariant prompt distribution constraint, while it is ignored in PGP.

In conclusion, we conduct a comprehensive analysis of prompt tuning for the consistency objective, which provides a complete guarantee to eliminate the interference of new tasks on previous tasks. As demonstrated in our ablation study in the Experiment section, the consistency of Aggregation and LayerNorm also contribute to reducing forgetting, and thereby they should not be ignored. We make a comparison of the performance between PGP and our approach in the next subsection.

### D.2  Performance Comparison

We compare with PGP [26] using the VPT-Seq and L2P [40] baselines on the four benchmarks in our experiments. The results are shown in Table 5. We implement PGP to VPT (*i.e.* VPT-PGP) under the same training settings as VPT-NSP$^2$ for a fair comparison. For the L2P-based methods, we insert prompts into the first three layers instead of only the first layer in the original implementation [40]. An orthogonal projection is also applied to the prompt pool which is essentially a linear layer in L2P-based models. We follow the training setting of PGP to train the L2P-based methods. The results in Table 5 demonstrate that our full approach can achieve more improvements in accuracy and reduce more forgetting than PGP. Even when applying only the projection matrix $\mathcal{B}_1$ for the Affinity operation, our approach also performs better than PGP, demonstrating the effectiveness of our proposed method for mitigating the interference problem.

Table 5: Comparison with PGP on four benchmarks and two continual learning baselines. "-$\mathcal{B}_1$" indicates only the projection matrix $\mathcal{B}_1$ is used in our approach

| Method | 10S-CIFAR-100 | | 20S-CIFAR-100 | | 10S-ImageNet-R | | 10S-DomainNet | |
|---|---|---|---|---|---|---|---|---|
| | Acc.↑ | Forgetting↓ | Acc.↑ | Forgetting↓ | Acc.↑ | Forgetting↓ | Acc.↑ | Forgetting↓ |
| VPT-Seq | 87.27 | 12.33 | 82.36 | 17.36 | 72.46 | 19.41 | 73.28 | 25.65 |
| VPT-PGP | 87.76 | 11.98 | 82.71 | 16.85 | 73.12 | 18.92 | 73.98 | 25.15 |
| VPT-NSP$^2$-$\mathcal{B}_1$ | 90.58 | 6.91 | 88.13 | 10.27 | 78.05 | 8.14 | 82.31 | 10.89 |
| VPT-NSP$^2$ | **91.74** | 3.28 | **89.89** | 4.91 | **78.88** | 5.06 | **83.54** | 8.54 |
| L2P | 84.12 | 6.36 | 81.46 | 8.69 | 61.25 | 9.32 | 65.73 | 10.19 |
| L2P-PGP | 84.70 | 5.96 | 82.04 | 8.11 | 62.01 | 8.55 | 66.31 | 9.63 |
| L2P-NSP$^2$-$\mathcal{B}_1$ | 86.39 | 4.60 | 82.99 | 7.34 | 64.10 | 7.17 | 67.48 | 8.21 |
| L2P-NSP$^2$ | **86.78** | 4.22 | **83.37** | 6.93 | **64.66** | 6.84 | **68.14** | 7.79 |

