# OpenReview forum: "Visual Prompt Tuning in Null Space for Continual Learning"
_NeurIPS.cc/2024/Conference — NeurIPS 2024 poster_

### Official Review · Reviewer_2TYW · 2024-07-05

**Soundness:** 4
**Presentation:** 4
**Contribution:** 4
**Rating:** 8
**Confidence:** 5

**Summary:**

This paper introduces the orthogonal projection into the visual prompt tuning for continual learning, which comprehensively considers the full operations of a transformer layer on the interference problem. Moreover, two sufficient consistency conditions for self-attention and an invariant prompt distribution constraint for LayerNorm are theoretically deduced, based on which an effective null-space-based approximation solution is introduced to implement the prompt gradient orthogonal projection for visual prompt tuning. Finally, extensive experimental results demonstrate the effectiveness of anti-forgetting on four class-incremental benchmarks with diverse pre-trained baseline models, and our approach achieves superior performances to state-of-the-art methods.

**Strengths:**

1)The research motivation for algorithms is reasonable, and the theoretical proof is solid. Constraining the new learnable parameter orthogonal to the previous weight is a reasonable motivation to prevent historical knowledge forgetting.

2)Extensive evaluation and amazing performance show the superiority of the proposed method.

**Weaknesses:**

1)Eq.(8) seems be error.

2)L139: Eq.(8) suggests that if the weight update ∆Θ is orthogonal to the previous input feature Xt during training in the new task, the corresponding output feature will remain unchanged. Moreover, will this degrade the model’s discriminative ability for the current task when the weight update is orthogonal to the previous feature?

3)What is the model’s complexity and running time compared to the baseline VPT?

4)Suggest using a more accurate diagram to describe the algorithm to help understand it more clearly.

**Questions:**

Please see “Weaknesses”

**Limitations:**

Yes

---

> ### Author Rebuttal · Authors · 2024-08-06
>
> **1:** The symbol "$\Rightarrow$"  may cause confusion in Eq. (8). Our intention was to convey that the left equation can be *simplified to yield* the right equation, rather than the left equation *being a sufficient condition* for the right one. We will correct this to ensure a more precise expression.
>
> **2:** The orthogonal projection will degrade the model’s discriminative ability for the current task, since the update direction is constrained into a subspace smaller than the original gradient space. This is a fundamental challenge known as stability-plasticity dilemma [48] in continual learning. The stability indicates the model’s discriminative ability for the old tasks, while the plasticity indicates the model’s discriminative ability for the current (new) task. A stronger stability usually causes a weaker plasticity and vice versa. **The plasticity-stability dilemma objectively exists in continual learning. We cannot completely eliminate this dilemma yet we can improve the overall accuracy by carefully balancing the stability and plasticity.**
>
> In our approach, two techniques help to enhance the plasticity of our model. **(1)** As introduced in the "Trade-off between Stability and Plasticity" section, we employ a hyper-parameter $\bar{\eta}$ to control the weight of each projection matrix: $\Delta \mathbf{P}=[\bar{\eta}\mathcal{B}_2 + (1-\bar{\eta})\mathbf{I}] \mathbf{P} _{\mathcal{G}} [\bar{\eta}\mathcal{B}_1 + (1-\bar{\eta})\mathbf{I}]$. When $\bar{\eta}$ is less than 1, the update direction of parameters is not strictly constrained to the orthogonal direction. Instead, the parameters can update in the original direction of the gradients. This relaxation enhances the model’s discriminative ability when learning a new task. As demonstrated in the experimental results of Figure 5, the model can achieve higher accuracy with higher (worse) forgetting as $\bar{\eta}$ decreases. It indicates the importance of the trade-off between stability and plasticity, and our approach can make a good trade-off. **(2)** As suggested in [36], the orthogonal projection matrix is constructed from an approximated null space, since an exact null space may not always exist in practice. The orthogonal projection can also be relaxed by the approximation to encourage the model to acquire new knowledge.
>
> > [48] Mermillod M, Bugaiska A, Bonin P. The stability-plasticity dilemma: Investigating the continuum from catastrophic forgetting to age-limited learning effects. Frontiers in psychology, 2013, 4: 504.
>
> **3:** (1) Complexity
>
> Compared to the baseline “VPT-Seq”, our approach introduces the following additional computation: 1) The null-space projections in each optimization step, *i.e.*, Eq. (25). For a group of prompts $\mathbf{P}$ of shape $M \times D$, the complexity of the two projections is $\mathcal{O}\left(DM(D+M)\right)$. Suppose there are $L$ ViT layers equipped with prompts. The batch size and epochs are represented as $n_{batch}$ and $n_{epoch}$, respectively. We denote the total number of samples in all $T$ tasks as $n_{total}$. After training all the tasks, the complexity introduced by the null-space projections is $\mathcal{O}\left( \frac{n_{total} n_{epoch} LDM(D+M)}{n_{batch}} \right)$. 2) The loss of prompt distribution. $i.e.$, Eq. (26). We use $E_{dist}$ to represent the computational overhead of the distribution loss between two scalar elements. Considering all the layers and optimization iterations during training, the complexity introduced by the prompt distribution loss is $\mathcal{O}\left( \frac{n_{total} n_{epoch} LDM E_{dist}}{n_{batch}} \right)$. 3) The forward process to obtain $\mathbf{Q} _{X_t} \mathbf{W}_k^\top$ and $\mathbf{S} _{P_t}$ for each sample after the training stage of a task, *i.e.*, line 21 of Algorithm 1 in the appendix. We denote the computational cost of the model’s forward propagation as $E _{model}$. Thereby, the introduced additional computation is $n _{total} E _{model}$. 4) Computation of uncentered covariance matrices, *i.e.*, line 24 of Algorithm 1. The complexity of computing the two uncentered covariance matrices is $\mathcal{O}\left( n _{total}^2 N^2 (D+M) \right)$ , where $N$=197 is the number of image tokens in the ViT-B/16 model. 5) Computation of the null-space projection matrices, *i.e.*, line 25 of Algorithm 1. In each task, the two projection matrices are only need to be computed and updated once. We use $E _{matrices}$ to denote the computational cost in this process. The total additional computation in $T$ tasks is $T E _{matrices}$.
>
> Overall, the complexity compared to the baseline VPT is $\mathcal{O}\left( \frac{n _{total} n _{epoch} LDM(D+M+E _{dist})}{n _{batch}} + n _{total} E _{model} + n _{total}^2 N^2 (D+M) + T E _{matrices} \right)$.
>
> (2) Running time
>
> We report the average running time over three runs for the baseline model and our approach on the four benchmarks in Table VI. Compared to the baseline VPT-Seq, the running time increases by 2\~9 minutes (2.38%\~6.43%) across these benchmarks, with an average increase of 4.5 minutes (3.84%). The additional running time introduced by our approach is acceptable as it constitutes only a small portion of the overall running time.
>
> **4:** Thank you for the valuable suggestion. We will add the diagram to describe our algorithm visually.

---

> > ### Comment · Area_Chair_m4X8 · 2024-08-10
> > **Please have a discussion**
> >
> > This paper receives mixed reviews. The authors have provided a detailed response. Please give your reply and check whether there is still unclear point for authors to clarify.

---

> > ### Comment · Reviewer_2TYW · 2024-08-11
> >
> > Thanks for the authors' response, that have addressed most of my concerns.

---

### Official Review · Reviewer_W3C1 · 2024-07-08

**Soundness:** 4
**Presentation:** 4
**Contribution:** 4
**Rating:** 7
**Confidence:** 5

**Summary:**

This paper introduces the orthogonal projection into the visual prompt tuning for continual learning, which comprehensively considers the full operations of a transformer layer on the interference problem. They propose two sufficient consistency conditions for the self-attention and an invariant prompt distribution constraint for LayerNorm, based on which an effective null-space-based approximation solution is introduced to implement the prompt gradient orthogonal projection for visual prompt tuning. Extensive experimental results show the superiority of the proposed method.

**Strengths:**

1)	The idea in this paper is novel and interesting. They introduce the orthogonal projection into the visual prompting tuning for continual learning, which comprehensively considers the full operations in the Transformer Layer.
2)	The experimental results are sufficient and superior to the SOTA methods.
3)	Overall, this paper is well organized and well-written, which is easy for the readers to follow.

**Weaknesses:**

1)	In figure 1 of the overall framework, it is unclear why the Qp term in the affinity matrix can be neglected. The authors should provide clear reasons for such conclusion.
2)	In the method part of Eq.(10), In order to satisfy F_{Z_t}=F_{Z_t-1}, the authors transform Eq.(10) into the following Eq.(11) and Eq.(12), which are two sufficient conditions. Why not optimizing Eq.(10) directly? Please give detailed reasons.
3)	What’s the main differences between the single-head and multi-head self-attention for introducing the proposed orthogonal projection into the visual prompting tuning for continual learning?
4)	In the method optimization part, the authors propose the approximation method as illustrated in Eq.(25). Why not using the optimization method proposed in PGP[26]? What are the main differences ?

**Questions:**

See the weaknesses above.

**Limitations:**

Yes

---

> ### Author Rebuttal · Authors · 2024-08-06
>
> **1:** In VPT-Deep [13], the output tokens corresponding to the input prompts of the current ViT layer will be replaced by new trainable prompts in the next ViT layer. Therefore, we do not need to compute the output tokens corresponding to the input prompts. According to the forward propagation of ViT, the output prompt tokens are derived from $\mathrm{Q}_P$. Consequently, the $\mathrm{Q}_P$ term can be neglected.
>
> Specifically, $\mathrm{Q}_P$ represents the prompt queries in the attention map. After the self-attention operation, the corresponding derived tokens of $\mathrm{Q}_P$ are denoted as $\mathrm{F} _{Q_P}$. Then $\mathrm{F} _{Q_P}$ will undergo another LayerNorm and an MLP to derive the output prompt tokens $\mathrm{Y} _{Q_P}$. $\mathrm{Y} _{Q_P}$ can be neglected due to the replacement of new trainable prompts. To reduce computation overhead, $\mathrm{Q}_P$ can be just neglected in the previous affinity matrix in self-attention. Note that omitting $\mathrm{Q}_P$ has no impact on the output image tokens of the ViT layer, as the subsequent Aggregation, LayerNorm and MLP operations are performed independently for each token.
>
> **2:** The reason is that optimizing Eq. (10) directly leads to difficulty in deriving the solution expressed in terms of $\Delta\mathbf{P}$. Eq. (10) introduces non-unique solutions and a quadratic term of $\Delta\mathbf{P}^2$, which is explained in detail as follows.
>
> According to Eq. (3), we derive the following equation when directly optimizing Eq. (10):
> $softmax\left(\frac{\begin{bmatrix}Q_{X_t}K_{X_t}^{\top}&Q_{X_t}K_{X_t}^{\top}\end{bmatrix}}{\sqrt{D}}\right)\begin{bmatrix}V_{X_t}\\\\ V_{X_t}\end{bmatrix}
> =softmax\left(\frac{\begin{bmatrix}Q_{X_t}K_{X_t}^{\top}&Q_{X_t}K_{X_{t+1}}^{\top}\end{bmatrix}}{\sqrt{D}}\right)\begin{bmatrix} V_{X_t}\\\\ V_{X_{t+1}}\end{bmatrix} $\
> It can be further expanded as:\
> $softmax\left(\frac{\begin{bmatrix}Q_{X_t}K_{X_t}^{\top}&Q_{X_t}[LN(P_t)W_k+b_k]^{\top} \end{bmatrix}}{\sqrt{D}}\right)\begin{bmatrix}V_{X_t}\\\\ LN(P_t)W_v+b_v\end{bmatrix}
> =softmax\left(\frac{\begin{bmatrix}Q_{X_t}K_{X_t}^{\top}&Q_{X_t}[LN(P_t+\Delta P)W_k+b_k]^{\top}\end{bmatrix}}{\sqrt{D}}\right)\begin{bmatrix}V_{X_t}\\\\ LN(P_t+\Delta P)W_v+b_v\end{bmatrix}$
>
> However, it is hard to simplify the above equation to derive a solution expressed in terms of $\Delta\mathbf{P}$. First, the non-injection property of the softmax function causes non-unique solutions. That is to say, we cannot derive $\mathbf{a}=\mathbf{b}$ from $softmax\left(\mathbf{a}\right)=softmax\left(\mathbf{b}\right)$. Second, when we omit the softmax operation, the multiplication between $\mathbf{Q}_{X_t}\left[LN\left(\mathbf{P}_t+\Delta\mathbf{P}\right)\mathbf{W}_k\right]^\top$ and $LN\left(\mathbf{P}_t+\Delta\mathbf{P}\right)W_v$ derives a quadratic term $LN\left(\mathbf{P}_t+\Delta\mathbf{P}\right)^\top LN\left(\mathbf{P}_t+\Delta \mathbf{P}\right)$, which results in difficult optimization. Due to the above obstacles, we transform Eq. (10) into Eq. (11) and Eq. (12), rather than optimizing Eq. (10) directly.
>
> **3:** The main difference is that the parameter update should be orthogonal to the subspace spanned by the concatenation matrices from all heads for multi-head self-attention. Specifically, for the singe-head self-attention, the parameter update should be orthogonal to the subspaces spanned be $\mathbf{Q} _{X_t} \mathbf{W}_k^\top$ and $\mathbf{S} _{P_t}$ according to Eq. (23) and Eq. (24). When considering the multi-head self-attention, the subspaces should be spanned by $[\mathbf{Q} _{X_t.1} \mathbf{W}_k^\top; \mathbf{Q} _{X_t.2} \mathbf{W}_k^\top; \cdots; \mathbf{Q} _{X_t.H} \mathbf{W}_k^\top]$ and $[\mathbf{S} _{P_t.1}; \mathbf{S} _{P_t.2}; \cdots; \mathbf{S} _{P_t.H}]$, where $\mathbf{Q} _{X_t.h}$ and $\mathbf{S} _{P_t.h}$ denote the corresponding intermediate activations in the $h$-th head ($h\in\\{1,2,⋯,H\\}$), respectively. $H$ is the number of heads, and "$[;]$" represents the concatenation of matrices along the first dimension. Therefore, only an additional step of concatenation of the corresponding matrices from all heads is required for introducing the proposed orthogonal projection into the multi-head self-attention.
>
> **4:** In PGP [26], the matrices to be optimized in their two conditions are element-wise summed. Then, SVD is applied to the summed matrix to obtain the orthogonal projection matrix. To enable the addition of the two matrices, PCA dimensionality reduction is also used on one of the matrices to align the dimensions.
>
> However, this summing approach is not applicable in our method, since the two matrices $\mathcal{B}_1$ and $\mathcal{B}_2$ are multiplied on the right and the left, respectively. Even with dimensionality reduction, they cannot be summed and merged into a single matrix. In fact, the two projection matrices have different meanings: $\mathcal{B}_1$ is a constraint on individual tokens, while $\mathcal{B}_2$ is a constraint on a specific dimension of all tokens at the dimension level. Therefore, it is not suitable to add them directly in terms of their respective meanings. Consequently, the optimization method used in PGP [26] cannot be applied in our case.

---

> ### Comment · Reviewer_W3C1 · 2024-08-09
>
> I appreciate your detailed response. The authors have addressed all the concerns I raised in my initial review, and after considering the other feedback, I kept my score.

---

### Official Review · Reviewer_4B73 · 2024-07-08

**Soundness:** 3
**Presentation:** 3
**Contribution:** 3
**Rating:** 6
**Confidence:** 5

**Summary:**

This work aims to eliminate the interference on previously learned knowledge for visual prompt tuning in the field of continual learning, so that catastrophic forgetting can be mitigated. To this end, it analyzes the conditions for keeping the output features unchanged in the transformer block that features the self-attention mechanism. Two consistency conditions for the self-attention are derived by deducing the proposed two sufficient conditions for the consistency objective. Moreover, a constraint on the distribution of the prompts is proposed to further simplify the LayerNorm operation. Consequently, the interference can be eliminated in theory by achieving the proposed two consistency conditions and the prompt-invariance constraint. The proposed approach implements the two consistency conditions by performing two null-space projections on the prompt gradients during training a new task; the constraint is implemented by an additional loss function that penalizes the drifting of prompt distribution across sequential tasks. Substantial experiments demonstrate the effectiveness of the proposed approach.

**Strengths:**

This work comprehensively analyzes the conditions for learning without interference in the transformer-based visual prompt tuning. The proposed two consistency conditions with the constraint provide a theoretical guarantee on eliminating the interference problem. With this solid mathematical support, the proposed approach, which performs null-space projection on prompt gradients, shows significant improvements in reducing forgetting and increasing accuracy. The effectiveness of the approach is validated on extensive experiments, involving four class-incremental benchmarks and various pre-training datasets and paradigms. By visualizing the evolution of training losses, the effectiveness in mitigating interference is demonstrated as well. The proposed approach also achieves state-of-the-art performance on the four benchmarks. Besides, the adaptive nullity and plasticity enhancement strategy is also well-motivated and validated to be an effective way of balancing stability and plasticity. The paper is clearly structured and easy to follow. The figures in the paper are clear and well-designed, enhancing the overall readability and comprehension of the paper.

**Weaknesses:**

1. The consistency objective Eq. (10) is decomposed into two sufficient conditions, namely Eq. (11) and Eq. (12). A detailed explanation is necessary to understand why direct analysis and simplification of Eq. (10) is not pursued.
2. The variable Q_p is omitted during the deduction of consistency conditions. Further explanation is needed to clarify why it can be disregarded.
3. In the Adam-NSCL [36] method, the trade-off between stability and plasticity is controlled solely by the nullity. In contrast, this approach involves adding weighted identity matrices to the projection matrices. How do these two methods differ in balancing stability and plasticity? A deeper discussion is expected to highlight the advantage of the proposed trade-off strategy.
4. The number of training epochs in this approach (100) is greater than that in other approaches (e.g., 50 epochs in DualPrompt for ImageNet-R). It would be better to provide a justification for this training setting.
5. As described in line 4 of Algorithm 2, the projection matrix is normalized by a Frobenius norm. There is a lack of explanation for this operation.

**Questions:**

In addition to the parameters of the prompts, the classifier contains parameters that require updates (i.e. the weights and biases) as well. Why not perform an orthogonal projection on the parameters of the classifier to reduce forgetting?

**Limitations:**

Reasonable discussion on limitations is included in the paper.

---

> ### Author Rebuttal · Authors · 2024-08-06
>
> **1:** We do not simplify Eq. (10) directly because it leads to difficulty in deriving the solution expressed in terms of $\Delta\mathbf{P}$. Eq. (10) introduces non-unique solutions and a quadratic term of $\Delta\mathbf{P}^2$, which is explained in detail as follows.
>
> According to Eq. (3), we derive the following equation when directly simplifying Eq. (10):
> $softmax\left(\frac{\begin{bmatrix}Q_{X_t}K_{X_t}^{\top}&Q_{X_t}K_{X_t}^{\top}\end{bmatrix} }{\sqrt{D}}\right)\begin{bmatrix}V_{X_t}\\\\V_{X_t}\end{bmatrix}
> =softmax\left(\frac{\begin{bmatrix}Q_{X_t}K_{X_t}^{\top}&Q_{X_t}K_{X_{t+1}}^{\top}\end{bmatrix}}{\sqrt{D}}\right)\begin{bmatrix} V_{X_t}\\\\ V_{X_{t+1}}\end{bmatrix} $\
> It can be further expanded as:\
> $softmax\left(\frac{\begin{bmatrix}Q_{X_t}K_{X_t}^{\top}&Q_{X_t}[LN(P_t)W_k+b_k]^{\top} \end{bmatrix}}{\sqrt{D}}\right)\begin{bmatrix}V_{X_t}\\\\ LN(P_t)W_v+b_v\end{bmatrix}
> =softmax\left(\frac{\begin{bmatrix}Q_{X_t}K_{X_t}^{\top}&Q_{X_t}[LN(P_t+\Delta P)W_k+b_k]^{\top}\end{bmatrix}}{\sqrt{D}}\right)\begin{bmatrix}V_{X_t}\\\\ LN(P_t+\Delta P)W_v+b_v\end{bmatrix}$
>
> However, it is hard to simplify the above equation to derive a solution expressed in terms of $\Delta\mathbf{P}$. First, the non-injection property of the softmax function causes non-unique solutions. That is to say, we cannot derive $\mathbf{a}=\mathbf{b}$ from $softmax\left(\mathbf{a}\right)=softmax\left(\mathbf{b}\right)$. Second, when we omit the softmax operation, the multiplication between $\mathbf{Q}_{X_t}\left[LN\left(\mathbf{P}_t+\Delta\mathbf{P}\right)\mathbf{W}_k\right]^\top$ and $LN\left(\mathbf{P}_t+\Delta\mathbf{P}\right)W_v$ derives a quadratic term $LN\left(\mathbf{P}_t+\Delta\mathbf{P}\right)^\top LN\left(\mathbf{P}_t+\Delta \mathbf{P}\right)$, which results in difficult simplification. Due to the above obstacles, we propose two sufficient conditions (*i.e.*, Eq. (11) and Eq. (12)), rather than simplifying Eq. (10) directly.
>
> **2:** In VPT-Deep [13], the output tokens corresponding to the input prompts of the current ViT layer will be replaced by new trainable prompts in the next ViT layer. Therefore, we do not need to compute the output tokens corresponding to the input prompts. According to the forward propagation of ViT, the output prompt tokens are derived from $\mathbf{Q}_P$. Consequently, the variable $\mathbf{Q}_P$ can be omitted.
>
> Specifically, $\mathbf{Q}_P$ represents the prompt queries in the attention map. After the self-attention operation, the corresponding derived tokens of $\mathbf{Q}_P$ are denoted as $\mathbf{F} _{Q_P}$.  Then $\mathbf{F} _{Q_P}$ will undergo another LayerNorm and an MLP to derive the output prompt tokens $\mathbf{Y} _{Q_P}$. $\mathbf{Y} _{Q_P}$ can be neglected due to the replacement of new trainable prompts. To reduce computation overhead, $\mathbf{Q}_P$ can be just neglected in the previous affinity operation in self-attention. Note that omitting $\mathbf{Q}_P$ has no impact on the output image tokens of the ViT layer, as the subsequent Aggregation, LayerNorm and MLP operations are performed independently for each token.
>
> **3:** The difference lies in that Adam-NSCL balances stability and plasticity solely by controlling the nullity, while our method first achieves a near-optimal stability point and then enhances plasticity by weakening the orthogonal constraints. Adding a weighted identity matrix to the gradients in our approach is more flexible for enhancing plasticity. In our method, the orthogonal constraint is strict when $\bar{\eta}=1$, and the model's stability reaches a near-optimal level. As $\bar{\eta}$ gradually decreases to 0, the orthogonal constraints are progressively relaxed until completely eliminated. This enables the model to achieve maximum plasticity for learning new tasks. Therefore, incorporating a weighted identity matrix into orthogonal constraints facilitates the model's efficient and effective acquisition of new knowledge.
>
> **4:** Orthogonal projection-based methods constrain the update direction of parameters during training. To fully optimize the model parameters and converge to a (local) optimal point, the model usually requires adequate training with more epochs. Therefore, we adopt a larger number of training epochs.
>
> **5:** The projection matrix is normalized by a Frobenius norm to provide an upper bound for the scale of gradients after projection. Specifically, the Frobenius norm is sub-multiplicative [47]. For two matrices $\mathbf{A}$ and $\mathbf{B}$, the inequality $||\mathbf{AB}||_F\le||\mathbf{A}||_F||\mathbf{B}||_F$ holds. For convenience, we use $\tilde{\mathbf{U}}_0$ to denote $\mathbf{U}_0 \mathbf{U}_0^\top$ in line 4 of Algorithm 2. Then the projection matrix is denoted as $\mathcal{B}=\frac{\tilde{\mathbf{U}}_0}{||\tilde{\mathbf{U}}_0||_F}$.  Consequently, when the gradient matrix $\mathbf{P} _\mathcal{G}$ is multiplied by the projection matrix $\mathcal{B}$, we have $ ||\mathcal{B}\mathbf{P} _\mathcal{G}||_F \le ||\mathcal{B}||_F ||\mathbf{P} _\mathcal{G}||_F=|\mathbf{P} _\mathcal{G}||_F$. This demonstrates that using the Frobenius norm can provide an upper bound for the scale of the projected gradients, thereby preventing excessive gradient magnitudes.
>
> > [47] Meyer C D. Matrix analysis and applied linear algebra Society for Industrial and Applied Mathematics, 2023.
>
> **6:** Response to the question: The reason is that the classifier in each task is trained independently rather than continuously. The classifiers of previously learned tasks remain fixed during training on new tasks, eliminating the need for gradient calculations. Moreover, the classifier of the current task is only updated in this task and is unaffected by the classes of previous tasks, rendering orthogonal constraints unnecessary. As a result, orthogonal projections are not required for classifiers in any task.

---

> > ### Comment · Area_Chair_m4X8 · 2024-08-10
> > **Please have a discussion**
> >
> > This paper receives mixed reviews. The authors have provided a detailed response. Please give your reply and check whether there is still unclear point for authors to clarify.

---

> > ### Comment · Reviewer_4B73 · 2024-08-14
> > **Rebuttal Response**
> >
> > I have read the rebuttals from authors and discussions from others. And most of my concerns have been addressed. Thus, I maintain my initial score. Thanks.

---

### Official Review · Reviewer_rV5K · 2024-07-13

**Soundness:** 3
**Presentation:** 2
**Contribution:** 3
**Rating:** 7
**Confidence:** 5

**Summary:**

This paper proposes a novel paradigm for continual learning based on prompt tuning. By deriving the constraints for orthogonal projection of prompt gradients in ViTs, the method aims to minimize forgetting during the learning process. Experiments on four benchmarks show that NSP² achieves superior performance by avoiding forgetting.

**Strengths:**

1. The paper focuses on orthogonal projection methods in continual learning, extending CNN-based methods to the ViT architecture, resulting in performance improvements.
2. The method is supported by mathematical derivations of the derived constraints, providing strong theoretical guarantees.
3. The paper addresses continual learning based on pre-trained models, which is a valuable topic.

**Weaknesses:**

1. While orthogonal gradient updates help prevent forgetting in continual learning, they also hinder positive knowledge transfer between tasks. In methods like L2P, similar tasks might select the same prompts, potentially improving performance on previous tasks after learning new ones.
2. The derivation process in the methods section is overly lengthy, which leads to a very brief description of the experimental section. The comparison with existing methods is reduced to just three lines of text. Consider moving some derivations to the appendix.
3. The experimental tables contain many blanks, which is not conducive to a systematic comparison of different methods' performance and weakens the contribution of this method.
4. The paper omits comparisons with some leading methods, such as DAP[1], which is also based on prompt tuning.

[1] Jung D, Han D, Bang J, et al. Generating instance-level prompts for rehearsal-free continual learning[C]. Proceedings of the IEEE/CVF International Conference on Computer Vision. 2023: 11847-11857.

**Questions:**

According to my understanding, as the number of tasks increases, the shrinking gradient subspace will make gradient updates increasingly difficult. Therefore, I would like to see the performance of NSP² on longer sequences of continual learning tasks.

**Limitations:**

As mentioned in the paper, additional constraints are introduced to simplify the consistency conditions.

---

> ### Author Rebuttal · Authors · 2024-08-07
>
> **1:** In theory, it seems that orthogonal projection methods do not have the merits of backward knowledge transfer. However, this problem can be alleviated by two techniques in our approach. **(1)** We adopt a plasticity enhancement strategy which employs a hyper-parameter $\bar{\eta}$ to control the weight of each projection matrix: $\Delta P=[\bar{\eta}B_2+(1-\bar{\eta})I]P_ \mathcal{G}[\bar{\eta}B_1+(1-\bar{\eta})I]$. By incorporating weighted identity matrices, the strict orthogonal update direction can be relaxed to some extent. This relaxation enhances the model’s ability to integrate new knowledge when learning a new task. **(2)** The orthogonal projection matrix is constructed from an approximated null space as suggested in [36], since an exact null space may not exist in practice. The orthogonal projection can also be relaxed by the approximation.
>
> We utilize the experiments on both **long-term** and **regular** CL settings to demonstrate the effectiveness of our approah.
>
> **(1)** We experiment on 5 benchmarks under the protocols of 50 tasks and 100 tasks to validate that our approach remains effective even within the context of **long-term CL**. The results are presented in **Table I** in the PDF and are strongly recommended for review by the reviewer. Despite lacking plasticity enhancement, VPT-NSP2 can outperform existing SOTA approaches and especially surpasses L2P by a large margin. This demonstrates that **forgetting is still the predominant factor affecting performance in long sequence of tasks**.
> With the plasticity enhancement, VPT-NSP2 achieves significant increase in accuracy (by 1.1%\~2.9%). **This demonstrates that our plasticity enhancement is effective in learning new knowledge in long-term CL.**
>
> **(2)** The experimental results across the four **regular CL** benchmarks are shown in Table II. NSP2 also outperforms L2P significantly even without plasticity enhancement. It achieves higher accuracy on all the benchmarks when using plasticity enhancement. Figure A shows the effects of the orthogonal projection weight $\bar{\eta}$. Accuracy can be improved with the decrease of $\bar{\eta}$, validating that **the proposed relaxed orthogonal constraints on gradients can promote learning new knowledge to achieve better performance**. The steady decrease in forgetting also verifies that **orthogonal constraints can be relaxed by decreasing $\bar{\eta}$**.
>
> In the above experiments, our approach focusing on anti-forgetting outperforms L2P and our baseline (i.e. VPT-Seq) significantly, implying that **forgetting has a greater impact on the performance of continual learning than backward knowledge transfer.**
> In L2P, prompts from old tasks may be selected and trained without constraints in new tasks, causing interference between tasks and leads to forgetting. Subsequent improvements, such as CODA-Prompt [32] and CPrompt [11], mitigate this issue by freezing prompts from old tasks and incrementally training new prompts for new tasks, thereby preventing the interference on old tasks and achieving better performance. The development of technical approaches also reflects that **catastrophic forgetting remains a predominant challenge to be addressed in continual learning**.
>
> Overall, the backward knowledge transfer ability is influenced by the plasticity of models. Nevertheless, the stability-plasticity dilemma remains **a fundamental challenge** in the field of CL. Achieving a better trade-off in this dilemma can enhance both anti-forgetting and backward knowledge transfer abilities. **In future work, we will focus on improving the backward transfer ability of our approach while maintaining its anti-forgetting capability to achieve better performance.**
>
> **2:** The detailed derivation process in the method section is crucial for explaining the derivation of the two projection matrices used in our approach. However, we understand the need for a more detailed experimental section. We appreciate the reviewer's suggestion and will consider moving some derivations to the appendix to provide more detailed experimental conclusions.
>
> **3:** We have made extensive efforts over the past week to reproduce 6 approaches for a more systematic comparison, including EvoPrompt [18] (AAAI’24), OVOR-Deep [12] (ICLR’24), DualP-PGP [26] (ICLR’24), InfLoRA [20] (CVPR’24), EASE [45] (CVPR’24) and CPrompt [11] (CVPR’24). The results are highlighted in blue in Table III. VPT-NSP2 achieves the highest accuracy, surpassing the second-best method by 0.6%\~2.4% across the four benchmarks.
>
> **4:** In Table III and Table 2 in our paper, we compare with 12 prompt-tuning-based methods proposed in 2023 and 2024. DAP has the problem of **batch information leakage which results in comparison unfairness.** As stated by Zhou et al. [46] in the discussions on comparison fairness, "during inference,... it is equal to directly annotating the task identity. When removing the batch information... a drastic degradation in the performance ..."
>
> The reproduced results reported in [46] are shown in Table IV. The accuracy declines by 4.8\~42.6% with an average of 27.3% when eliminating the batch information leakage. Besides, we reproduce DAP on four benchmarks used in our experiments. We also observe a drastic degradation (by 16.42%\~23.24%) in accuracy, as shown in Table V.
>
> The official code of DAP, line 210-236 in file vit.py, assumes that test samples are batched and all of them come from the same task, which is an unreasonable assumption.
>
> >[46] Zhou D W, et al. Continual learning with pre-trained models: A survey. preprint arXiv:2401.16386, 2024.
>
> **5:** Response to the question: The experimental results for 50 and 100 tasks across 5 benchmarks are shown in Table I. Our approach surpasses the second-best competitor by 0.4%~6.4% with an average improvement of 2.0%, demonstrating that **our approach has the ability to handle longer sequences of continual learning tasks**.

---

> > ### Comment · Area_Chair_m4X8 · 2024-08-10
> > **Please have a discussion**
> >
> > This paper receives mixed reviews. The authors have provided a detailed response. Please give your reply and check whether there is still unclear point for authors to clarify.

---

> ### Author Response · Authors · 2024-08-08
> **Concerns addressed**
>
> Dear reviewer rV5K,
>
> Thank you very much for reviewing our paper and giving us some good questions. We have tried our best to answer all the questions according to the comments. Especially, we conduct experiments on **long-term continual tasks** across 5 benchmarks, and compare our approach with 6 existing SOTA methods. **We eagerly expect the reviewer to examine the PDF for the results and comparison.** The experimental results in Table I show that our approach remains effective and outperforms SOTA methods including L2P under the long-term CL protocol. It demonstrates that our approach can mitigate the shrinking gradient subspace and learn new knowledge effectively by the proposed relaxion for orthogonal projections, even under the setting of longer sequences of continual learning tasks.
>
> We sincerely hope that our responses can address all your concerns. Is there anything that needs us to further clarify for the given concerns?
>
> Thank you again for your hard work.

---

> > ### Comment · Reviewer_rV5K · 2024-08-10
> > **Concerns partially addressed**
> >
> > I appreciate the detailed response from the authors and the efforts they made over the past week. The authors have partially addressed my concerns as follows:
> >
> > 1. The proposed method NSP² performs well on long-sequence CL tasks, and by adjusting the hyperparameter $\bar{\eta}$, it can further relax the orthogonality constraint to facilitate learning more new knowledge.
> > 2. More results from comparison methods have been presented, which enhances the credibility and contribution of the paper.
> >
> > However, I still have some concerns:
> >
> > 1. Continual learning should not just involve learning each task separately to prevent forgetting. Like human learning mechanisms, algorithms should perhaps focus more on how to leverage existing knowledge to quickly and accurately master a new skill, as well as how to use new knowledge to better solve previous tasks, rather than learning entirely separately, like separate multi-task learning. Nevertheless, the orthogonal prompt tuning method proposed by the authors for Vision Transformers also provides a solid theoretical foundation for applying ViTs to continual learning, which is a significant contribution.
> > 2. The results provided in Table III of the PDF seem to be lower for the comparison methods than those reported in the original papers, and the differences are substantial. For example, for 20S-CIFAR100, EASE is reported to achieve 91.51 in the original paper, while the authors report only 85.80. Similar discrepancies are observed for 10S-CIFAR100. I would like to understand if there are any differences in implementation or experimental settings.
> >
> > For these reasons, I have increased my score to 5. If the authors can further address my above concerns, I will consider raising the score further.

---

> > > ### Author Response · Authors · 2024-08-10
> > >
> > > Thank you for your kind reply and support for our work. We are glad that we have addressed most of your concerns. Below is our response to the remaining concerns in your comments.
> > >
> > > **Reply to Q1:** This is a valuable viewpoint on continual learning. It has always been our pursuit to enable deep neural networks to learn new knowledge better by utilizing the experience of previously learned knowledge like human beings. In our future work, we will consider exploring mining the parameters of different importance on learning old and new tasks, and combining this scheme with anti-forgetting techniques such as orthogonal projection to enhance knowledge transfer capabilities without forgetting.
> > >
> > > Specifically, the learnable parameters in a network can be divided into three parts according to the contribution to learning new knowledge and preserving old knowledge by the following process. After finishing training on an old task and before learning on a new task, we can use a criterion (e.g., gradients) to **measure the importance of different parameters on the old task and the new task**, respectively. Then we divide those parameters into three sets by: 1) **high importance on the old task**, 2) **high importance on the new task**, and 3) **moderate importance on both old and new tasks**. During training in the new task, different strategies are adopted for these three sets of parameters. 1) For the parameters important to the old task, we use **a completely strict orthogonal constraint** on them to reduce forgetting and keep stability. 2) For the parameters important to the new task, we **do not perform any constraint** on them to maximize the stability. 3) For the parameters of moderate importance on both old and new task, we use **an appropriate relaxed orthogonal constraint** on them. In this way, the whole model can be trained with an enhanced knowledge transfer capability and retain a good ability of anti-forgetting.
> > >
> > > **Reply to Q2:** Following L2P, the accuracies reported in Table III of our PDF for **ALL the methods** are the "**final average accuracy**" (*i.e.*, 85.80 instead of 91.51 for EASE). Therefore, we would like to emphasize that **the comparison in our table is fair.**
> > >
> > > Specifically, **EASE reports two different metrics** for accuracy in the original paper, which are referred to as "final average accuracy (FAA)" and "mean average accuracy (MAA)" here. They correspond to $\bar{\mathcal{A}}$ and $\mathcal{A}_B$ in the Table 1 of EASE, respectively.
> > >
> > > The finally average accuracy represents the accuracy over all learned tasks when the model finishes continual training on the last task. It is defined as:
> > >
> > > $\mathcal{A} _B = \frac{1}{B} \sum _{i=1} ^{B} a _{B,i}$
> > >
> > > where $B$ is used to denote the number of total tasks to correspond to the symbols used in EASE, and $a_{B,i}$ is the accuracy of the $B$-th model (*i.e.*, the model after training on the last task) on the $i$-th task’s test data.
> > >
> > > The mean average accuracy is the mean value of $\mathcal{A} _1, \mathcal{A} _2, \cdots, \mathcal{A} _B$:
> > >
> > > $\bar{\mathcal{A}}=\frac{1}{B}\sum_{i=1}^{B}\mathcal{A}_i$
> > >
> > > Since the accuracy in early learning tasks are usually high under the class-incremental learning protocol, *i.e.*, $\mathcal{A}_1 > \mathcal{A}_2 > \cdots > \mathcal{A}_B$ usually holds, MAA is almost always higher than FAA. This explains why the results in the column $\bar{\mathcal{A}}$ are higher than those in the column $\mathcal{A}_B$ in the Table 1 of EASE. For a fair comparison with other methods, we report FAA in Table III which are widely adopted in existing prompt-tuning-based papers.
> > >
> > > For a clearer comparison **under these two metrics**, we select those methods which report both FAA and MAA from Table III of the PDF, including: InfLoRA [20], EASE [45] and CPrompt [11]. The results are shown in **Table VII**. **The columns of FAA have been reported in Table III (corresponding to "Acc."), and the columns of MAA are newly added for a comparison under the mean average accuracy metric.** It can be seen that our approach surpasses other approaches in both metrics. Especially, **VPT-NSP2 outperforms EASE by an average of 3.86% under the FAA metric and an average of 3.35% under the MAA metric.**
> > >
> > > **Table VII**: Comparison with the methods that report both finally average accuracy (FAA) and mean average accuracy (MAA). **The italic values are produced by us due to lack of official results, while the others are from their corresponding original papers.** Best results are highlighted in bold.
> > > ||20S-CIFAR-100|20S-CIFAR-100|10S-CIFAR-100|10S-CIFAR-100|10S-ImageNet-R|10S-ImageNet-R|10S-DomainNet|10S-DomainNet|
> > > |:-:|:-:|:-:|:-:|:-:|:-:|:-:|-:|:-:|
> > > |Method|FAA|MAA|FAA|MAA|FAA|MAA|FAA|MAA|
> > > |EASE|85.80|91.51|87.76|92.35|76.17|81.73|*78.89*|*84.58*|
> > > |InfLoRA|*81.42*|*87.42*|87.06|91.59|75.65|80.82|*81.45*|*88.75*|
> > > |CPrompt|*83.97*|*90.08*|87.82|92.53|77.14|82.92|82.97|88.54|
> > > |VPT-NSP2|**89.89**|**93.75**|**91.74**|**96.02**|**78.88**|**84.84**|**83.54**|**88.94**|

---

> > > > ### Comment · Reviewer_rV5K · 2024-08-12
> > > > **Concerns addressed**
> > > >
> > > > Thanks for your detailed explanation. Overall, NSP² performs very well, and it would be even better if further exploration could be made in promoting knowledge transfer between tasks. The authors have addressed almost all of my concerns, so I have raised my score to 7.

---

> ### Author Response · Authors · 2024-08-12
> **Concerns addressed**
>
> Dear Reviewer rV5K,
>
> We sincerely thank you very much for the meaningful comments and providing the valuable feedback. We really hope that our responses can address all the remaining concerns. Thank you again for your great help and many good questions and suggestions, which largely help improve the quality of our paper.
>
> We would like to clarify if you have further concerns. Thanks very much.

---

### Author Rebuttal · Authors · 2024-08-07

We thank all reviewers for their valuable feedback, with three reviewers (W3C1, 2TYW and 4B73) strongly supporting our work. We are encouraged that reviews think our paper:
- **the idea in this paper is novel and interesting** (by Reviewer W3C1);
- **the theoretical proof is solid** (by Reviewer 2TYW);
- **providing strong theoretical guarantees** (by Reviewer rV5K);
- **substantial experiments demonstrate the effectiveness of the proposed approach** (by Reviewer 4B73).

The main concern of reviewer rV5K is that orthogonal constraints may hinder learning new knowledge and degrade the performance on long-term continual learning. We address this issue by **conducting experiments with respect to 50 and 100 tasks across 5 benchmarks**. The results demonstrate that our approach remains **effective and superior** even within the context of long-term continual learning. Moreover, our experimental analysis on regular CL benchmarks also verifies that orthogonal constraints **can be relaxed by the proposed plasticity enhancement**, which **promotes the model to learn new knowledge and achieve better performance**.

All questions are addressed in reviewer-specific responses. Additionally, please find the PDF attached with helper tables and figures. These are referenced and described in our individual responses to reviewers.

---

### Decision · Program_Chairs · 2024-09-25

**Decision:**

Accept (poster)

**Comment:**

This paper receives unanimous accept recommendations. After a careful check of paper, comments, and rebuttal, the AC agrees with the reviewers and think the authors has well addressed the reviewer concern. The idea of this paper is novel and the result is good for continual learning. Thus, the AC makes a accept recommendation.